# LeanDojo: Theorem Proving with Retrieval-Augmented Language Models

**Kaiyu Yang**[1], **Aidan M. Swope**[2], **Alex Gu**[3], **Rahul Chalamala**[1], **Peiyang Song**[4],
**Shixing Yu**[5], **Saad Godil**,[*] **Ryan Prenger**[2], **Anima Anandkumar**[1,2]
[1]Caltech, [2]NVIDIA, [3]MIT, [4]UC Santa Barbara, [5]UT Austin
`https://leandojo.org`

## Abstract

Large language models (LLMs) have shown promise in proving formal theorems using proof assistants such as Lean. However, existing methods are difficult to reproduce or build on, due to private code, data, and large compute requirements. This has created substantial barriers to research on machine learning methods for theorem proving. This paper removes these barriers by introducing *LeanDojo*: an open-source Lean playground consisting of toolkits, data, models, and benchmarks. LeanDojo extracts data from Lean and enables interaction with the proof environment programmatically. It contains fine-grained annotations of premises in proofs, providing valuable data for *premise selection*—a key bottleneck in theorem proving. Using this data, we develop *ReProver* (Retrieval-Augmented Prover): an LLM-based prover augmented with retrieval for selecting premises from a vast math library. It is inexpensive and needs only one GPU week of training. Our retriever leverages LeanDojo's program analysis capability to identify accessible premises and hard negative examples, which makes retrieval much more effective. Furthermore, we construct a new benchmark consisting of 98,734 theorems and proofs extracted from Lean's math library. It features challenging data split requiring the prover to generalize to theorems relying on novel premises that are never used in training. We use this benchmark for training and evaluation, and experimental results demonstrate the effectiveness of ReProver over non-retrieval baselines and GPT-4. We thus provide the first set of open-source LLM-based theorem provers without any proprietary datasets and release it under a permissive MIT license to facilitate further research.

## 1 Introduction

Reasoning is a cornerstone of human intelligence and a fundamental goal of AI [3]. One prominent task is automated theorem proving (ATP): automatically generating proofs for theorems expressed in formal logic. ATP is useful for formal mathematics, producing mathematical proofs that can be checked rigorously [4]. Furthermore, it underpins formal verification, which is essential for proving the correctness and safety of high-stakes applications [5, 6].

ATP is challenging since the search space is prohibitively large. In many applications, it is impractical to generate proofs fully automatically. Therefore, interactive theorem proving (ITP) has emerged as an alternative paradigm. In ITP, proofs are constructed by human experts interacting with software tools called proof assistants, such as Coq [7], Isabelle [8], and Lean [1]. Machine learning can automate such interactive theorem proving, opening up a new avenue for theorem proving [9]. The model can learn to interact with proof assistants, given data containing human-written proofs.

---

[*]Research conducted while Saad Godil was at NVIDIA.

37th Conference on Neural Information Processing Systems (NeurIPS 2023) Track on Datasets and Benchmarks.

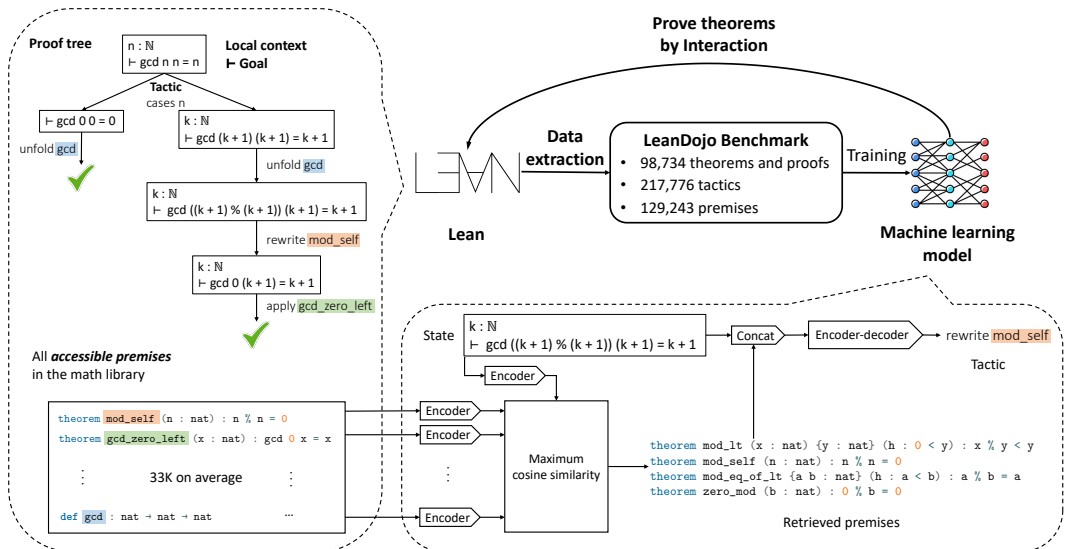

Figure 1: **Top right**: LeanDojo extracts proofs in Lean [1] into datasets for training machine learning models. It also enables the trained model to prove theorems by interacting with Lean's proof environment. **Top left**: The proof tree of a Lean theorem $\forall n \in \mathbb{N}$, `gcd n n = n`, where `gcd` is the greatest common divisor (details in Sec. 3). When proving the theorem, we start from the original theorem as the initial state (the root) and repeatedly apply tactics (the edges) to decompose states into simpler sub-states, until all states are solved (the leaf nodes). Tactics may rely on premises such as `mod_self` and `gcd_zero_left` defined in a large math library. E.g., `mod_self` is an existing theorem $\forall n \in \mathbb{N}$, `n % n = 0` used in the proof to simplify the goal. **Bottom**: Our ReProver model (Sec. 5). Given a state, it retrieves premises from the math library, which are concatenated with the state and fed into an encoder-decoder Transformer [2] to generate the next tactic.

Formal theorem proving serves as an important challenge for machine learning. From a computer science perspective, formal proofs can be treated as programs [10]. But unlike conventional programs in C++ or Python, the correctness of proofs can be verified using proof assistants. Therefore, theorem proving may be considered a special form of code generation, with rigorous evaluation and no room for the model to hallucinate. This can be consequential to current large language models (LLMs), as they have demonstrated exceptional capability in code generation [11] but have flaws in factuality and hallucination [12]. In addition, augmenting LLMs with external tools, such as proof assistants, has shown promise in improving their various capabilities, including multi-step reasoning [13].

Current research on LLMs for theorem proving is facing many barriers. To our knowledge, none of the existing LLM-based provers are open-source [14–21]. They all use private pretraining data, and the compute requirements can reach thousands of GPU days [17]. Furthermore, some rely on tailored infrastructure for distributed training and interaction with the proof assistant—both are not possible to fully reproduce without open-source code [17, 19]. We change the status quo by introducing LeanDojo: open-source toolkits, models, and benchmarks that give researchers access to state-of-the-art LLM-based provers with modest computational costs.

**Tools for Data Extraction and Interaction.** We focus on Lean, a proof assistant popular among mathematicians.[2] Our framework LeanDojo provides two essential functions for learning-based theorem proving (Fig. 1): extracting data and enabling models to interact with Lean programmatically.

For data extraction, LeanDojo extracts training data not directly visible in the raw Lean code (Fig. 2), e.g., proof trees consisting of intermediate states between proof steps (Fig. 1 *Top left*). In addition, LeanDojo is the first tool to locate premises in Lean proofs, enabling training machine learning models for premise selection. For interaction, LeanDojo turns Lean into a gym-like interactive environment [22]. Using LeanDojo, the model can observe proof states, change the state by executing

---

[2]"Lean" in our paper refers to Lean 3 by default. Lean 4 is not backward-compatible but is also supported by LeanDojo. Our Lean 4 results are in Appendix D.

proof steps (referred to as "tactics" in proof assistants), and receive feedback from Lean. LeanDojo is the first tool capable of interacting with Lean reliably, reducing proof-checking errors in existing tools [19] (correct proofs misjudged as incorrect) from 21.1% to 1.4%.

**Retrieval-Augmented LLMs for Theorem Proving.** LeanDojo addresses a key bottleneck in theorem proving: *premise selection* [23, 24]. Existing LLM-based provers generate the next proof step (tactic), taking only the current state as input. However, proving theorems depends critically on the premises, such as lemmas and definitions, from a math library.

For example, Fig. 1 (*Top left*) illustrates the proof of "$\forall n \in \mathbb{N}$, `gcd n n = n`", where gcd stands for greatest common divisor. The proof starts from the original theorem as the initial state and repeatedly applies tactics to decompose states into simpler sub-states, until all states are solved. Tactics may rely on premises such as `mod_self` and `gcd_zero_left` defined in a large math library. E.g., `mod_self` is an existing theorem "$\forall n \in \mathbb{N}$, `n % n = 0`" useful for simplifying the goal.

Incorporating all possible premises is too large to fit into LLMs' input, given the limited context window. Existing methods must learn to memorize the association between the proof state and the name `mod_self`. It works if the premise has been used in the training data to solve similar goals, but does not generalize to truly novel scenarios, e.g., theorems requiring lemmas unseen in training.

One potential solution is to complement memorization with explicit premise selection. LeanDojo extracts premise data from Lean, including where they are defined and used. It enables us to tackle premise selection by augmenting LLMs with retrieval. We introduce *ReProver* (Retrieval-Augmented Prover) (Fig. 1 *Bottom*): Given the current state, it generates a tactic conditioning on a small number of premises retrieved from Lean's math library, `mathlib` [25].

We need to limit retrieval to a small number of premises for it to be effective, and ideally, they should contain the ground truth premise. Our retriever builds upon Dense Passage Retriever (DPR) [26] but incorporates two algorithmic innovations: First, not all premises are accessible when proving a theorem (Sec. 3). LeanDojo can perform program analysis on Lean code to determine accessible premises. On our data, that reduces the average number of premises from 128K to 33K, significantly simplifying the retriever's task. Second, DPR needs negative examples in training and benefits from hard negatives, i.e., irrelevant premises that are hard to distinguish from ground truth ones. We propose in-file negatives: a simple mechanism to find hard negatives in premise selection, which samples negative premises defined in the same Lean source file as the ground truth premise.

**LeanDojo Benchmark.** Using LeanDojo, we construct a benchmark containing 98,734 theorems/proofs extracted from `mathlib`. Our benchmark is one of the largest math-focused theorem-proving datasets. We find that the common practice of splitting theorems randomly into training/testing has led to an overestimated performance in the previous papers. LLMs can prove seemingly difficult theorems simply by memorizing the proofs of similar theorems during training. In LeanDojo Benchmark, we mitigate this issue by designing challenging data split requiring the model to generalize to theorems relying on novel premises that are never used in training.

We use LeanDojo Benchmark to train and evaluate ReProver. Training takes only five days on a single GPU. In evaluation, ReProver can prove 51.2% theorems, outperforming a baseline that generates tactics directly without retrieval (47.6%) and another baseline using GPT-4 [27] to generate tactics in a zero-shot manner (29.0%). We also test ReProver on two existing datasets, MiniF2F [28] and ProofNet [29]. It can prove 26.5% theorems in MiniF2F and 13.8% in ProofNet, which is competitive with state-of-the-art methods without reinforcement learning [19], even though trained using far fewer resources. Moreover, it can prove 65 theorems that currently do not have proofs in Lean. Thus, our tool can also serve as an effective tool for augmenting existing math libraries in Lean.

**Contributions.** In summary, we make four main contributions: First, we introduce tools for extracting data from and interacting with Lean. Second, we develop ReProver, the first retrieval-augmented language model for theorem proving. Third, we construct a challenging benchmark for learning-based theorem proving and use it to validate the effectiveness of ReProver. Finally, we facilitate open research on LLMs for theorem proving by releasing our data, model, and code. Our method does not rely on private datasets and can be trained on a single GPU within a week. We believe this will significantly lower the barriers to academic research in this area and establish the first accessible baselines for future work to build upon. Further, our method can be used to automatically generate new Lean proofs without requiring human effort.

## 2 Related Work

**Theorem Proving.** Classical provers express theorems in first-order logic and search for proofs automatically in a large space [30, 31]. Even with data-driven search heuristics [32, 33], they fail to scale to large formalization projects. Therefore, recent work on learning-based theorem proving has focused on an alternative paradigm: automating the interaction with proof assistants.

The architecture of learning-based provers progressed from classical machine learning algorithms such as KNN [34], to graph neural networks explicitly encoding the syntax of formal expressions [9, 35], and now Transformer-based LLMs treating expressions as plain strings [14]. Besides the model architecture, researchers have explored several complementary dimensions: proof search algorithms for assembling model-generated steps into complete proofs [17, 21]; overcoming data scarcity through reinforcement learning (RL) [17, 19, 36, 37] or synthetic/auxiliary data [16, 38–40]; as well as outsourcing some proof goals to classical provers [18, 41–43]. Our base model without retrieval is a combination of straightforward design choices. It generates tactics by finetuning an encoder-decoder Transformer, ByT5 [44], via supervised learning without RL or auxiliary data. Then it searches for proofs using best-first search. Our model's algorithmic novelty lies in the retrieval.

**Premise Selection.** Selecting useful premises is recognized as a key challenge in theorem proving [23, 24, 45, 46]. Machine learning methods for premise selection have also progressed from classical models [41, 47, 48], recurrent neural networks [24], graph neural networks [38], to Transformers [49, 50]. However, existing methods either tackle premise selection in isolation without theorem proving [24, 38, 48] or feed the premises to a symbolic prover [41, 47, 49]. To our knowledge, we are the first to augment a learning-based formal theorem prover with retrieved premises so that the prover can learn how to use them effectively. For example, it can decide whether to use an explicitly retrieved premise or an implicitly memorized one.

**Data and Tools for Theorem Proving.** Tools for data extraction and interacting with proof assistants have been crucial drivers of learning-based theorem proving. Existing tools and datasets can be divided by proof assistants: Coq has GamePad [51], CoqGym [9], and PRISM [52]; Isabelle has IsarStep [53] and PISA [15]; HOL Light has HOList [54] and HoLStep [55], and Lean has LeanStep [16] and `lean-gym` [19]. MiniF2F [28] is the only cross-system dataset, with 488 theorems for evaluation. However, it does not have training theorems and is restricted to the domain of math olympiads.

Among available tools extracting data from proof assistants, LeanDojo is the only one that can extract premises for retrieval-augmented theorem proving. A few existing datasets also have premises [49, 54], but their data extraction tools are not public, making it difficult to construct new datasets. In addition, LeanDojo is the only tool that can interact with Lean robustly (Sec. 4) and can extract data from Lean 4. See Appendix A.3 for a detailed comparison between LeanDojo and alternatives.

**Mathematical Reasoning in Natural Language.** We focus on proving theorems expressed in formal logic, whereas researchers have also produced a plethora of work on mathematical reasoning in natural language [56–63]. A particularly relevant task is autoformalization, translating natural language texts into formal theorems and proofs [29, 64–72].

**Retrieval-Augmented Language Models.** Our ReProver is the first retrieval-augmented language model for formal theorem proving, though similar architectures have been studied extensively in NLP [73–81]. In addition, there have been many retrieval-augmented methods for code generation [82–88]. Most of them retrieve from a corpus not directly related to the current file, e.g., GitHub or Stack Overflow. In contrast, our retrieval corpus consists of premises accessible to the current file, which is determined by program analysis using LeanDojo. This is similar to what CoCoMIC [88] does for Python. However, their retrieval is based on heuristics, whereas ours is learned.

## 3 Background: Theorem Proving in Lean

At a high level, Lean is a programming language that allows you to write not only conventional programs but also theorems and proofs. To that end, it provides two pieces of machinery: First, it provides a unified language for defining programs, mathematical objects, theorems, and proofs, based on functional programming with dependent types [89]. Second, it provides a tactic system for constructing machine-checkable proofs semi-automatically.

```
data/nat/lemmas.lean

theorem mod_self (n : nat) : n % n = 0 :=
begin
  rw [mod_eq_sub_mod (le_refl _), nat.sub_self, zero_mod]
end
```

```
data/nat/gcd.lean

def gcd : nat → nat → nat                    -- gcd z y
| 0        y := y                            -- Case 1: z == 0
| (x + 1) y := gcd (y % (x + 1)) (x + 1)     -- Case 2: z > 0

theorem gcd_zero_left (x : nat) : gcd 0 x = x := begin simp [gcd] end

theorem gcd_self (n : nat) : gcd n n = n :=
begin
  cases n,
  { unfold gcd },
  unfold gcd,
  rewrite mod_self,
  apply gcd_zero_left
end
```

Import

Figure 2: Definition of greatest common divisor (`gcd`) in Lean and two related theorems. The proof of `gcd_self` (between "begin" and "end") relies on a premise `mod_self` imported from another file in the math library. Lean can run this proof to produce the proof tree in Fig.1 (*Top left*).

We use a simple example in Fig. 2 to illustrate how theorems are formalized and proved in Lean.[3] Here we want to formalize the greatest common divisor (`gcd`) of two natural numbers. First, we define `gcd` as a recursive function, taking two natural numbers as parameters and returning their `gcd` via the Euclidean algorithm. Then, we state a lemma named `gcd_zero_left` that $\forall x \in \mathbb{N}$, `gcd 0 x = x`, which can be proved simply by the definition of `gcd`. Finally, we state our main theorem `gcd_self` that $\forall n \in \mathbb{N}$, `gcd n n = n`, followed by its proof consisting of five tactics. In theorem proving, we are only concerned with generating the proof, i.e., the part between "begin" and "end"; everything before "begin" is known, including other files imported.

The syntax of tactics is quite expressive. They can take arguments and can be combined into compound tactics. You can think of tactics as programs in a domain-specific language (DSL). Users can extend the DSL by defining new tactics. This discrete, combinatorial, and unbounded action space makes theorem proving challenging for machine learning.

Another challenge is premise selection. Premises are existing lemmas or definitions useful for proving a theorem. They are used as arguments in tactics. For example, in Fig. 2 and Fig. 1 (*Top left*), the tactic "`rewrite mod_self`" rewrites the goal using the premise `mod_self`, which is defined in another file imported by the current file. Proofs cannot use premises that haven't been defined. For example, `gcd_self` cannot be used to prove `gcd_zero_left`. In addition, they cannot use premises not imported to the current file. Still, premises come from a large math library containing hundreds of thousands of existing definitions and theorems, making it hard, for humans and machines alike, to select the right premises when generating a tactic. This is a key bottleneck in theorem proving and is what we aim to address through retrieval-augmented LLMs.

## 4   LeanDojo: Toolkit and Benchmark

LeanDojo serves two essential needs of learning-based theorem proving in Lean. First, it extracts training data from Lean, and we use this capability to construct a challenging theorem proving benchmark. Second, it enables the model to interact with Lean programmatically.

**Data Extraction.**   Lean repos (e.g., `mathlib` or `lean-liquid`) contain source code of human-written theorems/proofs. However, the raw code is unsuitable for training the prover. It lacks runtime information that humans can access when using Lean, such as intermediate states between proof steps. Therefore, LeanDojo extracts the following information not directly visible in the code:

---

[3]The process is similar in many other proof assistants, though they may have different logical foundations.

- *File dependencies and abstract syntax trees (ASTs):* LeanDojo processes the repo to produce a directed acyclic graph whose nodes are files and edges are import relations between files. In addition, LeanDojo produces the AST of each file. File dependencies and ASTs are useful for program analysis, e.g., collecting theorems defined in a file or premises accessible to a theorem.

- *States and tactics:* LeanDojo extracts all tactics in proofs. For each tactic, it also extracts the states before/after the tactic, which allows us to reconstruct the proof tree in Fig. 1 (*Top left*).

- *Premises:* For each premise, such as `mod_self` in Fig. 2, LeanDojo records where it is defined (location in `data/nat/lemma.lean`) and where it is used (locations across many files). In addition, premises have unique fully qualified names (e.g., `nat.mod_self`) but are often used by ambiguous short names (`mod_self`), relying on Lean to perform name resolution. LeanDojo is capable of recording their full names.

Lean has basic support for exporting dependencies, ASTs, states, and tactics. However, it cannot resolve the premises' full names and locate their definitions. Therefore, we modify Lean to record this information (details in Appendix A.1). The modified Lean is used only for data extraction but not for evaluation, so we do not risk accidentally breaking Lean's logical soundness.

**LeanDojo Benchmark.** We construct a benchmark for premise selection and theorem proving, named *LeanDojo Benchmark*. The data is extracted from mathlib,[4] Lean's centralized math library covering diverse topics such as analysis, algebra, and geometry.[5] LeanDojo Benchmark is one of the largest math-focused theorem proving datasets, consisting of 98,734 theorems from 3,384 Lean files. Unlike existing datasets in Lean [16], LeanDojo Benchmark also contains the definitions of 130,262 premises, including not only theorems but also other definitions that can be used as premises (e.g., gcd in Fig. 2. Furthermore, the dataset has 217,776 tactics, 129,243 of them with at least one premise. The average number of premises is 2.13 among tactics with premises. Appendix B contains additional information on data format, datasheet [90], hosting, and licensing.

src/algebra/quaternion.lean
```
lemma conj_mul : (a * b).conj = b.conj * a.conj := begin
  ext; simp; ring_exp
end

lemma conj_conj_mul : (a.conj * b).conj = b.conj * a := begin
  rw [conj_mul, conj_conj]
end

lemma conj_mul_conj : (a * b.conj).conj = b * a.conj := begin
  rw [conj_mul, conj_conj]
end
```

Figure 3: Similar theorems/proofs are common. If splitting them randomly into training/testing, the model can prove testing theorems by memorization.

LeanDojo Benchmark has 94,734/2,000/2,000 theorems for training/validation/testing. It features a challenging data split for testing the prover's generalization in more realistic scenarios. Splitting theorems randomly can overestimate the prover's performance, by allowing it to prove many theorems through memorization. In human-written Lean code, a common idiom is to have a block of similar theorems/proofs for slightly different properties of the same math concept. For example, in Fig. 3, the last two theorems not only look similar but have identical proofs. If one of them is in training, the model can easily prove the other one by memorization. This shortcut enables the model to prove seemingly nontrivial theorems, including those requiring premises to prove.

To mitigate this issue, besides the `random` split, we create a challenging data split named `novel_premises`. It requires testing proofs to use at least one premise that has never been used in training. For example, the last two theorems in Fig. 3 both use the premise conj_mul. If one theorem is in the training set of the `novel_premises` split, the other one must also be in training.

---

[4]We use the commit `19c869efa56bbb8b500f2724c0b77261edbfa28c` released on October 11, 2023.

[5]More details, statistics, and visualizations of `mathlib` can be found at `https://leanprover-community.github.io/mathlib_stats.html`.

**Interacting with Lean.** Another important function of LeanDojo is to interact with Lean programmatically. It turns Lean into a gym-like environment [22], in which the prover can observe the proof state, run tactics to change the state, and receive feedback on errors or on proof completion. This environment is indispensable for evaluating/deploying the prover or training it through RL.

Below is LeanDojo's main interface for interacting with Lean through tactics. Lean also supports other proof styles not based on tactics. Although we only support tactic-style proofs, they are sufficiently general since any proof can be converted to a tactic-style proof.[6]

- `initialize(theorem)`: Given the theorem to prove, LeanDojo returns the initial state. A valid state is a string representing current proof goals and local contexts (see the nodes in Fig. 1 *Top left*). When there are multiple goals, their strings are concatenated.
- `run_tac(state, tactic)`: Run a tactic on a given state and return the next state. The returned state will be an error state if the tactic execution is not successful, e.g., due to timeout or inapplicable tactic. If the input state is an error, the result can only be an error.

Building this environment is technically challenging, as Lean is designed for human users, not machines. LeanDojo is the first tool that can interact with Lean reliably. Existing tool [19] is limited: 21.1% of the ground truth proofs are misjudged as incorrect, due to issues with how they construct the proof environment, which distorts the reported performance and produces unreliable feedback when used in reinforcement learning. In contrast, LeanDojo reduces the number of misjudgments to 1.4%. Details are in Appendix A.2.

## 5  ReProver: Retrieval-Augmented Theorem Prover

We develop the ReProver model that uses retrieval to select premises explicitly. At its core is a retrieval-augmented tactic generator (Fig. 1 *Bottom*). Given the current proof state, it retrieves a handful of potentially useful premises and generates a tactic conditioning on the concatenation of the state and retrieved premises. When proving theorems, the model generates multiple tactic candidates at each step, which are used in a standard best-first search algorithm to find proofs [16, 18, 19, 28].

**Premise Retrieval.** Our retriever is based on Dense Passage Retriever [26]. Given a state $s$ as the query and a library of candidate premises $\mathcal{P} = \{p_i\}_{i=1}^N$, it retrieves a ranked list of $m$ premises $\{p_i'\}_{i=1}^m$ from $\mathcal{P}$. In DPR, $s$ and $p_i$ are both raw texts but are embedded in a vector space, and we retrieve the top $m$ premises maximizing the cosine similarity between the state and the premise.

More formally, we have a function $f$ parameterized by $\theta$ for embedding both the state and the premises into a $h$-dimensional vector space: $f(s, \theta), f(p_i, \theta) \in \mathbb{R}^h$. We retrieve premises maximizing $f(s, \theta)^T f(p_i, \theta)/(\|f(s, \theta)\|_2 \|f(p_i, \theta)\|_2)$. We choose $f$ to be a Transformer encoder [2] followed by average pooling: $f(\cdot, \theta) = \text{AvgPool}(\text{Enc}(\cdot, \theta))$.

The retrieval is efficient. The premise embeddings $f(p_i, \theta)$ can be pre-computed, and we only need one forward pass to compute $f(s, \theta)$. We do not rerank the retrieved premises as in Magnushammer [49], which is more costly since it requires a separate forward pass for each retrieved premise.

Similar to DPR, we train the retriever by minimizing a contrastive loss between positive premises and in-batch negative premises. Specifically, suppose we have a batch of $b$ states. For each state, we sample a positive premise from the ground truth and $n$ negative premises from $\mathcal{P}$.[7] They are called "in-batch" negatives because they are shared by all states in the batch—Every state is associated with all $b \cdot (n+1)$ premises; at least 1 of them is positive. Let $l_{ij} \in \{0, 1\}$ denote whether a state-premise pair $(s_i, p_j)$ is positive. We minimize the mean squared loss:

$$\mathcal{L}(\theta) = \sum_{i=1}^{b} \sum_{j=1}^{b \cdot (n+1)} \left| l_{ij} - \frac{f(s_i, \theta)^T f(p_j, \theta)}{\|f(s_i, \theta)\|_2 \|f(p_j, \theta)\|_2} \right|^2. \tag{1}$$

---

[6]Another common type of proofs is "term-style proofs". Any term-style proof "`X`" can always be converted into an equivalent tactic-style proof "`exact X`", though such conversion may lead to unidiomatic proofs.

[7]When training the retriever, we ignore proof states followed by tactics without using any premise.

**Retrieving from Accessible Premises.** We incorporate into DPR two insights tailored to premise selection. First, instead of retrieving from all premises in the math library, we restrict to premises accessible to the current theorem. They include premises defined in the same file before the theorem, as well as those imported from other files. We compute accessible premises for each theorem, relying on LeanDojo's capability in program analysis (Sec. 4). Focusing on accessible premises makes $\mathcal{P}$ much smaller. LeanDojo Benchmark contains 130,262 premises in total, but the average number of accessible premises is only 33,160.

**In-file Negative Examples.** DPR's performance depends critically on the quality of negative examples [91, 92]. In early experiments, we sampled all $n$ negative premises randomly, and the model often mistakenly retrieved other premises from the same file as the positive one. Therefore, we propose a scheme that samples $k$ in-file negatives and $n - k$ random negatives for training.

**Tactic Generation.** As in Fig. 1 (*Bottom*), retrieved premises are concatenated with the state.[8] Then an encoder-decoder Transformer, ByT5 [44], takes them as input and generates the tactic. The model is trained to minimize the cross entropy loss w.r.t. human-written tactics.

Training ReProver takes substantially less compute than prior methods (120 GPU hours vs. more than 1000 hours [16, 17]). All existing LLM-based provers pretrain on datasets specific to math and coding [14–20]. The pretraining is computationally expensive, and the datasets are kept private. In contrast, we choose to avoid domain-specific pretraining and build upon `google/byt5-small`—a model checkpoint that is generic, publicly available, and relatively small (299M parameters vs. 837M [16] or 600M [17]). We could see further benefits from domain-specific pretraining, as in Minerva [57], or stronger LLMs like LLaMA [93] or StarCoder [94], but that is beyond our scope. In addition, our model is finetuned on human-written tactics only, without auxiliary data [16] or data collected through online interaction with Lean [17, 19]. These orthogonal directions are valuable but will significantly increase the method's complexity and compute requirements.

# 6 Experiments

We evaluate ReProver on LeanDojo Benchmark. It outperforms baselines on premise selection and theorem proving, demonstrating the promise of theorem proving with retrieval-augmented language models. Experimental details and hyperparameters are in Appendix C.1.

**Premise Selection.** For premise selection, we only use tactics in LeanDojo Benchmark that have at least one premise. The model, based on a ByT5 encoder, uses the state before a tactic as the query to retrieve 100 premises. Then, we calculate standard metrics in information retrieval: R@k (recall for the top $k$ retrieved premises) and MRR (mean reciprocal rank).

Our first baseline is a classical BM25 retriever [95] without machine learning. Results in Table 1 show that our method outperforms BM25 significantly across the board. However, it exhibits a large performance degradation on the challenging data split (comparing `novel_premises` to `random`). This is consistent with the general observation that machine learning can be brittle in the presence of distribution shifts. In addition, we compare with two ablations: one retrieving from all premises (instead of accessible premises only) and the other without in-file negatives. They perform worse than our method, demonstrating the effectiveness of our two improvements upon DPR.

**Theorem Proving Experimental Setup.** Then we evaluate ReProver on theorem proving. The training has two stages: First, we train the retriever and use it to retrieve 100 premises for all proof states in LeanDojo Benchmark. Second, we train the tactic generator, taking as input the concatenation of the state and retrieved premises (truncated to a length limit). During evaluation, the tactic generator is combined with best-first search to prove theorems. We evaluate the *Pass@1* metric: The prover is given only one attempt and must find the proof within a wall time limit of 10 minutes. Training takes five days on a single NVIDIA A100 GPU with 80GB memory, and evaluation takes two days on eight V100 GPUs. Please see Appendix C.1 for details.

**Baselines.** Following prior work [16, 28], we include `tidy` as a baseline. It is a tactic in `mathlib` that tries to complete the proof using heuristics (without machine learning). We apply `tidy` directly

---

[8]We retrieve 100 premises, concatenate them with the state, and truncate the concatenation to a fixed length.

Table 1: Premise selection testing performance. For each method, we train and evaluate two models independently using different data splits (`random` and `novel_premises`; see Sec. 4). R@k is the recall for the top $k$ retrieved premises, and MRR is the mean reciprocal rank metric (higher is better). Our retriever outperforms BM25 and ablations. Results for Lean 4 are in Appendix D.

| Method | random | | | novel_premises | | |
|---|---|---|---|---|---|---|
| | R@1 | R@10 | MRR | R@1 | R@10 | MRR |
| BM25 | 6.7 | 17.2 | 0.15 | 5.9 | 15.5 | 0.14 |
| w/ all premises | 1.9 | 11.9 | 0.08 | 2.1 | 12.4 | 0.08 |
| Ours | **13.5** | **38.4** | **0.31** | **9.1** | **27.6** | **0.24** |
| w/ all premises | 11.7 | 36.2 | 0.27 | 7.1 | 23.1 | 0.20 |
| w/o in-file negatives | 10.8 | 33.1 | 0.25 | 7.9 | 25.7 | 0.22 |

to the original theorem and see if it can succeed within the wall time limit. Another baseline uses GPT-4 as the tactic generator. Given a state, it queries GPT-4 to generate 35 tactics in zero-shot. After removing invalid ones, the remaining tactics are combined with best-first search to find proofs. Data contamination is possible: Many proofs had been publicly available on GitHub before GPT-4's data cutoff date (September 2021). See Appendix C.2 for details.

Unfortunately, it is not feasible to compare with existing LLM-based provers in Lean [16, 17, 19]. None of them are open-source or can be reproduced with reasonable effort. Furthermore, we cannot compare directly with the numbers reported in their papers, due to differences in data, infrastructure, and training procedures (details in Appendix C.3). Many difficulties are due to the private nature of existing methods. By releasing our code and models, we hope to create accessible baselines for future work to build upon.

Table 2: Theorem proving Pass@1 (%) on the testing data of LeanDojo Benchmark. Our ReProver model outperforms `tidy`, GPT-4, and a baseline that generates tactics directly without retrieval. Results for Lean 4 are in Appendix D.

| Method | random | novel_premises |
|---|---|---|
| `tidy` | 23.8 | 5.3 |
| GPT-4 | 29.0 | 7.4 |
| ReProver (ours) | **51.2** | **26.3** |
| w/o retrieval | 47.6 | 23.2 |

**Results.** Table 2 shows the results on the testing data of LeanDojo Benchmark. ReProver outperforms all baselines on two different data splits, demonstrating the effectiveness of retrieval-augmented theorem proving. GPT-4 performs substantially worse than our method, even though it may have seen the ground truth proofs due to data contamination. The task cannot be solved out of the box by state-of-the-art LLMs, calling for algorithmic innovations to make further progress.

Testing theorems in `novel_premises` are indeed much more challenging. All methods in Table 2 perform substantially worse on `novel_premises` than the `random` split. We argue that performance on challenging splits is more indicative of the prover's capability and should be emphasized in the future development of theorem proving.

**Evaluation on MiniF2F and ProofNet.** We run ReProver to prove theorems in MiniF2F [28] and ProofNet [29]. These two datasets are for testing only and do not have training theorems, which makes them challenging since the distribution of theorems is quite different from `mathlib` used to train ReProver. MiniF2F focuses on math olympiads, and ProofNet focuses on exercises in undergraduate math textbooks. On MiniF2F's test set in Lean, ReProver achieves a Pass@1 of 26.5%, which is competitive with state-of-the-art methods without RL (25.9% in Polu et al. [19]). On ProofNet, our Pass@1 is 13.8%, which is the first reported theorem proving result on this dataset. Further, many theorems do not have ground truth proofs in Lean. Our prover discovers 33 proofs in MiniF2F and 39 proofs in ProofNet that currently do not have Lean proofs. Please see Appendix C.4 for details, examples, and caveats.

# 7   Conclusion

We have introduced LeanDojo: an open-source playground for learning-based theorem proving in Lean, consisting of toolkits, models, and benchmarks. It extracts data from Lean and enables the model to interact with Lean programmatically. We have developed ReProver, the first retrieval-augmented LLM for theorem proving. Limitations and future work are discussed in Appendix F.

We have released our code, data, models, and documentation to facilitate future research:

- LeanDojo's codebase for data extraction and interaction with Lean: `https://github.com/lean-dojo/LeanDojo`
- LeanDojo's documentation: `https://leandojo.readthedocs.io`
- Datasets: (1) LeanDojo Benchmark: `https://doi.org/10.5281/zenodo.8016385` with DOI 10.5281/zenodo.8016385. (2) LeanDojo Benchmark 4 (Appendix D): `https://doi.org/10.5281/zenodo.8040109` with DOI 10.5281/zenodo.8040109.
- ReProver's code and models: `https://github.com/lean-dojo/ReProver`
- ChatGPT plugin (Appendix E): `https://github.com/lean-dojo/LeanDojoChatGPT`
- LeanDojo Website: `https://leandojo.org`

## Acknowledgments and Disclosure of Funding

This work is partially supported by Caltech's Center for Autonomous Systems and Technologies. Kaiyu Yang is supported by the Computing, Data, and Society Postdoctoral Fellowship at Caltech. Alex Gu is supported by the National Science Foundation (NSF) Graduate Research Fellowship. Rahul Chalamala and Peiyang Song are supported by the Summer Undergraduate Research Fellowships (SURF) program at Caltech. Anima Anandkumar is partially supported by the Bren endowed chair. We appreciate the valuable feedback from Logan Murphy and members of the Anima AI+Science Lab on an initial version of this paper. We thank Junyan Xu for manually inspecting the proofs generated by our model on ProofNet. We also thank Jeremy Avigad and Mario Carneiro for insightful discussions on supporting Lean 4 in LeanDojo.

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

# Appendix

# A  LeanDojo Technical Details

We provide more information on how LeanDojo extracts data from and interacts with Lean.[9] For further details, please check our open-source implementation.

## A.1  Extracting Premise Information from Lean's Elaborator

"Premises" in this paper belong to a category of Lean expressions called "constants." In Lean, definitions of constants are grouped into nested, hierarchical namespaces. Therefore, each premise has a unique fully-qualified name. For example, `mod_self` in Fig. 2 is defined in the namespace `nat`; therefore, its fully qualified name is `nat.mod_self`. However, it would be too verbose if premises had to be referred to using full names. In practice, tactics often refer to premises using short names such as `mod_self`. In case multiple premises share the same short name, Lean automatically infers the correct one from the context through a process called "name resolution". LeanDojo is able to trace the input/output of Lean's name resolution and thereby extract accurate premise information for training the retriever.

Name resolution in Lean is implemented in a process called "elaboration," which happens after parsing but before the parsed expressions are checked by Lean's trusted kernel. Elaboration takes as input user-entered expressions (called "pre-expressions") that are concise, partially specified, and potentially ambiguous. It turns them into complete expressions ready to be checked by the kernel. This is realized by inferring not only full names but also missing types, implicit arguments, overloading, type coercion, etc. Please refer to de Moura et al. [96] for details on Lean's elaboration process. In LeanDojo, we modify Lean's internal implementation, intercepting the elaborator to record its input/output:

- *Pre-expression*: The input to Lean's elaborator, including where premises are used in proofs.
- *Expression*: The output of the elaborator, including the premise's full name and where it is defined.

Locations are spans in the source code, specified by the file name and the row/column numbers of its start/end. Our modification takes the form of a Git patch that LeanDojo can automatically apply to any version of Lean 3 after March 24, 2022.

## A.2  Reliable Interaction with Lean

Polu et al. [19] introduced `lean-gym`. To our knowledge, it is the only mature, open-source tool before LeanDojo for interacting with Lean programmatically. However, we found severe issues with `lean-gym`: About 21.1% of the correct, human-written proofs are misjudged as incorrect, leading to two problems: First, it underestimates the prover's evaluation performance. Second, the results are too noisy as feedback signals for reinforcement learning.

After carefully analyzing `lean-gym`'s implementation, we identified the root cause of the problem. When proving a theorem, the environment used by `lean-gym` is subtly different from the original environment used by humans. Specifically, `lean-gym` fails to handle namespaces correctly (illustrated in Fig. A). As a result, name resolution fails unexpectedly when checking correct proofs.

For example, Fig. A compares the correct environment and the environment constructed by `lean-gym`. The theorem should be inside the namespace "`buffer`". However, in `lean-gym`, it merely opens the namespace. These two scenarios are different when it comes to name resolution. Being inside a namespace instructs Lean to favor constants defined in that namespace, whereas opening a namespace does not have such an effect. In this example, the short name "`read`" is ambiguous: We have "`monad_reader.read`" defined in "`init/control/reader.lean`" and "`buffer.read`" defined in "`data/buffer.lean`". In the correct environment, the "`read`" in "`unfold read`" resolves to "`buffer.read`". Whereas in `lean-gym`'s environment, it incorrectly resolved to "`monad_reader.read`". Lean complains that "`read`" is not an equational lemma, because it is referring to a wrong "`read`". LeanDojo does not suffer from this kind of error since it uses a different mechanism for constructing the environment. Specifically, it wraps the interaction code as a Lean tactic, which is inserted into the proof. Therefore, the environment is guaranteed to be correct.

---

[9]"Lean" in our paper refers to Lean 3 by default. Lean 4 is not backward-compatible but is also supported by LeanDojo. Our Lean 4 results are in Appendix D.

We quantitatively compare `lean-gym` and LeanDojo on the number of proof checking errors. In this study, we use Lean v3.42.1 paired with `mathlib` version 6e5ca7d0097313e59f7533a42e3ea5197484c775 since they are supported by both tools. We use LeanDojo to extract all tactic-style proofs and enter them into both tools. These proofs are all correct, but `lean-gym` failed on 21.1% of them. In contrast, LeanDojo only failed on 1.4%, and its failures are a subset of `lean-gym`'s. We include this study in our open-source repo and document example proofs from the remaining 1.4% to provide transparency on LeanDojo's limitations.[10]

```
import data.buffer                              import data.buffer
universe u                                      universe u

namespace buffer                                open buffer

theorem my_read_eq_read' {a : Type u} [inhabited a]   theorem my_read_eq_read' {a : Type u} [inhabited a]
(b : buffer a) (i : nat) (h : i < b.size) :     (b : buffer a) (i : nat) (h : i < b.size) :
  read b ⟨i, h⟩ = read' b i := begin              read b ⟨i, h⟩ = read' b i := begin
  cases b,                                        cases b,
  unfold read,                                    unfold read,
  unfold read',                                   unfold read',
  simp [array.read_eq_read']                      simp [array.read_eq_read']
end                                             end

end buffer
```

**Correct environment**                     **lean-gym's environment**

**ERROR: unfold tactic failed, `read` does not have equational lemmas nor is a projection**

Figure A: An example of correct proofs misjudged as incorrect by `lean-gym`, adapted from the theorem `read_eq_read'` in "`data/buffer.lean`" of Lean's standard library. The error message is because `lean-gym` failed to resolve the short name "`read`" to the correct fully-qualified name. The Lean code in this figure is only for illustrative purposes. It does not reflect the implementation technique used by `lean-gym` to construct the environment. Instead of generating actual Lean code, `lean-gym` uses Lean's metaprogramming APIs to construct the environment.

### A.3 Comparison with Existing Tools for Learning-Based Theorem Proving in Lean

To our knowledge, LeanStep [16][11] and `lean-gym` [19] are the only published tools for learning-based theorem proving in Lean. There are a few unpublished prototypes, such as repl, lean-client-python, and `lean-gym` for Lean 4, none of which is mature enough or is under active development. Therefore, we only compare LeanDojo with LeanStep and `lean-gym` (summarized in Table A).

**Functionality.** LeanDojo supports both data extraction and interacting with Lean programmatically. In contrast, LeanStep is only for data extraction, and `lean-gym` is only for interacting with Lean. They are not actively maintained, so they do not support recent versions of `mathlib` (tested on August 11, 2023, using `mathlib` commit 19c869efa56bbb8b500f2724c0b77261edbfa28c). Also, neither of them support Lean 4 (Appendix D). LeanDojo fully supports recent mathlib and Lean 4. Furthermore, LeanStep cannot extract premise information and is not applicable to repos other than `mathlib`. Last, LeanDojo comes with comprehensive documentation and unit tests, whereas other tools barely have any.

**Implementation details.** LeanStep and LeanDojo use different mechanisms to extract ASTs and proof trees. LeanStep implements an ad-hoc parser in Python for parsing Lean code into ASTs. It also intercepts Lean's tactic system to insert logging code. Then the logs are used to reconstruct proof trees. This implementation is brittle and does not work for the current versions of Lean/mathlib. In contrast, LeanDojo relies on Lean's built-in mechanisms for exporting ASTs and proof states (`lean --ast --tsast --tspp`), which works robustly for recent Lean/`mathlib`. This mechanism was developed after LeanStep.

---

[10]https://github.com/lean-dojo/LeanDojo/blob/main/tests/interaction/test_unexpected_errors.py

[11]LeanStep is technically a dataset. We are referring to the lean_proof_recording tool for extracting it.

Regarding interaction with Lean, both `lean-gym` and LeanDojo rely on Lean's metaprogramming APIs, and LeanDojo partially builds upon `lean-gym`'s code. However, `lean-gym` has a critical issue in that it misjudges many correct proofs as incorrect (Appendix A.2). The main reason is that `lean-gym` fails to distinguish two subtly different cases when constructing the proof environment: (1) opening a namespace; (2) being inside a namespace. LeanDojo does not suffer from this issue. Instead of operating as a standalone program in the IO monad, it wraps the interaction code into a special tactic, which is inserted into the correct location in the proof. Therefore, the interaction code is guaranteed to run in the same environment as the original human-written proof.

| | | LeanStep [16] | `lean-gym` [19] | LeanDojo (ours) |
|---|---|---|---|---|
| Data extraction | Premise information | ✗ | N/A | ✓ |
| | Lean 4 support | ✗ | N/A | ✓ |
| | Recent `mathlib` | ✗ | N/A | ✓ |
| | Repos other than `mathlib` | ✗ | N/A | ✓ |
| Interaction | Estimated errors | N/A | 21.1% | 1.4% |
| | Lean 4 support | N/A | ✗ | ✓ |
| | Recent `mathlib` | N/A | ✗ | ✓ |
| | Repos other than `mathlib` | N/A | ✓ | ✓ |
| | Documentation & unit tests | ✗ | ✗ | ✓ |

Table A: Comparing LeanDojo with existing tools for data extraction and interaction with Lean.

# B  LeanDojo Benchmark

## B.1  Dataset Format

We describe the data format of LeanDojo Benchmark, which has the following directory structure:

```
/
├── corpus.jsonl ............. All premises defined in mathlib and Lean's standard library
├── metadata.json .................................................... Metadata
├── licenses
│   ├── lean .................................... Attribution to Lean's Apache 2.0 license
│   ├── mathlib ............................. Attribution to mathlib's Apache 2.0 license
│   └── README.md ........ Statement that LeanDojo Benchmark is released under CC BY 2.0
├── random ....................................... Theorems/proofs of the random split
│   ├── train.json ............................................... 94,734 theorems
│   ├── val.json ................................................... 2,000 theorems
│   └── test.json .................................................. 2,000 theorems
└── novel_premises ..................... Theorems/proofs of the novel_premises split
    ├── train.json ............................................... 94,734 theorems
    ├── val.json ................................................... 2,000 theorems
    └── test.json .................................................. 2,000 theorems
```

**Premise Definitions.**  `corpus.jsonl` contains the definition of premises. It has 3,280 lines. Each line is in JSON format and corresponds to a Lean file. Below is an example for "init/control/functor.lean", which directly imports three other files: "init/core.lean", "init/function.lean", and "init/meta/name.lean". It defines two constants that can be used as premises: "functor" and "functor.map_const_rev". For each premise, we have access to its full name, the source code, and its start/end location within the file.

```
"path": "_target/deps/lean/library/init/control/functor.lean",
"imports": [
  "_target/deps/lean/library/init/core.lean",
  "_target/deps/lean/library/init/function.lean",
  "_target/deps/lean/library/init/meta/name.lean"
],
"premises": [
```

```
  {
    "full_name": "functor",
    "code": "class functor (f : Type u → Type v) : Type (max (
        u+1) v) :=\n(map : Π {α β : Type u}, (α → β) → f α →
        f β)\n(map_const : Π {α β : Type u}, α → f β → f α :=
        λ α β, map ∘ const β)",
    "start": [11, 1],
    "end": [13, 70],
    "kind": "class"
  },
  {
    "full_name": "functor.map_const_rev",
    "code": "@[reducible] def functor.map_const_rev {f : Type u
        → Type v} [functor f] {α β : Type u} : f β → α → f α
        :=\nλ a b, b <$ a",
    "start": [18, 1],
    "end": [19, 14],
    "kind": "definition"
  }
]
```

**Theorems and Tactics.** Theorems in LeanDojo Benchmark are split into training/validation/testing using two different strategies (Sec. 4). They are formatted in JSON, and below is an example corresponding to the theorem "`real.angle.to_real_pi_div_two`". LeanDojo has recorded two tactics: "`split`" and "`linarith [pi_pos]`". For each tactic, we have the proof states before/after it. The "`linarith [pi_pos]`" tactic illustrates how premises are recorded: They are annotated using HTML-like strings such as "`linarith [<a>pi_pos</a>]`", followed by a "provenance list". Each element in the list corresponds to a premise in the tactic.

```
"url": "https://github.com/leanprover-community/mathlib",
"commit": "19c869efa56bbb8b500f2724c0b77261edbfa28c",
"file_path": "src/analysis/special_functions/trigonometric/
    angle.lean",
"full_name": "real.angle.to_real_pi_div_two",
"start": [512, 9],
"end": [513, 56],
"traced_tactics": [
  {
    "tactic": "split",
    "annotated_tactic": ["split", []],
    "state_before": "⊢ -π < π / 2 ∧ π / 2 ≤ π",
    "state_after": "2 goals\n⊢ -π < π / 2\n\n⊢ π / 2 ≤ π"
  },
  {
    "tactic": "linarith [pi_pos]",
    "annotated_tactic": [
      "linarith [<a>pi_pos</a>]",
      [
        {
          "full_name": "real.pi_pos",
          "def_path": "src/analysis/special_functions/
              trigonometric/basic.lean",
          "def_pos": [122, 7],
        }
      ]
    ],
    "state_before": "⊢ -π < π / 2",
    "state_after": "no goals"
  }
```

```
]
```

Not all theorems have tactic-style proofs. For those without tactic-style proofs, concatenating the tactics does not lead to a complete proof of the original theorem. However, this is not an issue when using the data for theorem proving evaluation or for training tactic generators.

## B.2  Datasheet

We present a datasheet [90] for documentation and responsible usage of LeanDojo Benchmark.

**Motivation.**

- *For what purpose was the dataset created?* It was created as a benchmark for learning-based theorem proving in Lean.
- *Who created the dataset (e.g., which team, research group) and on behalf of which entity (e.g., company, institution, organization)?* It was created by the authors of this paper.
- *Who funded the creation of the dataset?* See the acknowledgments in Sec. 7.

**Composition.**

- *What do the instances that comprise the dataset represent (e.g., documents, photos, people, countries)?* The dataset consists of formal definitions, theorems, and proofs written in Lean [1].
- *How many instances are there in total (of each type, if appropriate)?* The dataset has 98,734 theorems and their proofs, as well as 130,262 premises defined in 3,384 files.
- *Does the dataset contain all possible instances or is it a sample (not necessarily random) of instances from a larger set?* The dataset contains all theorems/proofs that LeanDojo can extract from the commit `19c869efa56bbb8b500f2724c0b77261edbfa28c` of `mathlib` released on October 11, 2023.
- *What data does each instance consist of?* Theorems/proofs in the dataset are Lean code written by programmers and mathematicians.
- *Are relationships between individual instances made explicit?* Definitions in the dataset are linked to proofs using them as premises.
- *Are there recommended data splits?* Yes, we recommend two data splits: `random` and `novel_premises`. Please see Sec. 4 for details.
- *Are there any errors, sources of noise, or redundancies in the dataset?* ASTs extracted by LeanDojo contain a small number of errors due to potential flaws in Lean's AST exporting mechanism. However, they do not have a tangible impact on our work.
- *Is the dataset self-contained, or does it link to or otherwise rely on external resources (e.g., websites, tweets, other datasets)?* The dataset is self-contained.
- *Does the dataset contain data that might be considered confidential (e.g., data that is protected by legal privilege or by doctor-patient confidentiality, data that includes the content of individuals' non-public communications)?* No.
- *Does the dataset contain data that, if viewed directly, might be offensive, insulting, threatening, or might otherwise cause anxiety?* No.

**Collection Process.**

- *How was the data associated with each instance acquired?* The data is directly observable by opening `mathlib` in VS Code with the Lean plugin. However, we had to instrument Lean to export the data programmatically.
- *What mechanisms or procedures were used to collect the data (e.g., hardware apparatuses or sensors, manual human curation, software programs, software APIs)?* The data was generated by building a Lean repo using our modified Lean and postprocessing the exported data.

- *Who was involved in the data collection process (e.g., students, crowd workers, contractors), and how were they compensated (e.g., how much were crowd workers paid)?* No manual effort was involved in the data collection process.
- *Over what timeframe was the data collected?* The final version of the dataset was generated in October 2023.

**Uses.**

- *Has the dataset been used for any tasks already?* We have used the dataset for training and evaluating machine learning models on the tasks of premise selection and theorem proving.
- *Is there a repository that links to any or all papers or systems that use the dataset?* Yes, `https://leandojo.org`.

**Distribution.**

- *Will the dataset be distributed to third parties outside of the entity (e.g., company, institution, organization) on behalf of which the dataset was created?* Yes, the dataset is publicly available on the Internet.
- *How will the dataset be distributed (e.g., tarball on website, API, GitHub)?* The dataset can be downloaded as a tarball.
- *Will the dataset be distributed under a copyright or other intellectual property (IP) license, and/or under applicable terms of use (ToU)?* The dataset is distributed under CC BY 2.0. The data generation code is distributed under the MIT license. The dataset was extracted from mathlib, which depends on lean. Both of them are distributed under the Apache 2.0 license. We include their licenses in the dataset as attribution (Appendix B.1).
- *Have any third parties imposed IP-based or other restrictions on the data associated with the instances?* No.
- *Do any export controls or other regulatory restrictions apply to the dataset or to individual instances?* No.

**Maintenance.**

- *Who will be supporting/hosting/maintaining the dataset?* The authors of this paper.
- *How can the owner/curator/manager of the dataset be contacted (e.g., email address)?* Please contact Kaiyu Yang at `kaiyuy@caltech.edu`.
- *Is there an erratum?* No.
- *Will the dataset be updated (e.g., to correct labeling errors, add new instances, delete instances)?* Please check `https://leandojo.org` for any update.
- *If others want to extend/augment/build on/contribute to the dataset, is there a mechanism for them to do so?* Yes, they can use our data generation code, which is publicly available.

### B.3 Data Hosting, Licensing, and Maintenance

LeanDojo Benchmark is distributed under the CC BY 2.0 license. The data is hosted on zenodo.org (a long-term data repository operated by CERN). The LeanDojo tool for data extraction and interaction with Lean is released at `https://github.com/lean-dojo/LeanDojo` under the MIT license. Our model checkpoints are hosted on Hugging Face Hub. LeanDojo's documentation is hosted on Read the Docs at `https://leandojo.readthedocs.io`. LeanDojo's website (`https://leandojo.org`) is the entry point for everything related to it, including any future updates or maintenance.

## C  Experiments

### C.1  Details and Hyperparameters

The premise retriever and tactic generator in ReProver are initialized by the `google/byt5-small` checkpoint on Hugging Face. It is a T5-like [97] encoder-decoder Transformer that operates directly

on UTF-8 bytes without tokenization. We choose ByT5 [44] instead of T5 because Lean code makes extensive use of Unicode math symbols, which may cause problems to T5's pretrained tokenizer. The retriever uses the encoder only, whereas the generator uses both the encoder and the decoder.

In training, we use one NVIDIA A100 GPU with 80GB of memory. The code is implemented in PyTorch and PyTorch Lightning, with bfloat16 mixed precision and DeepSpeed ZeRO Stage 2 [98]. Both the retriever and the generator are optimized using AdamW [99] with a batch size of 8. In the first 2,000 steps, the learning rate warms up linearly from 0 to the maximum value. Then it decays to 0 following a cosine schedule. The maximum learning rate is $10^{-4}$ for the retriever and $5 \times 10^{-4}$ for the generator. When training the retriever, we sample 3 negative premises for each example, including 1 in-file negative premise. When training the generator, we apply dropout to retrieved premises with a dropout rate of 0.5. Then, we truncate the generator's input to 2,300 tokens.

During evaluation, the tactic generator is combined with best-first search to find proofs. At each search step, it produces 64 tactic candidates using beam search. Each tactic is associated with a log-likelihood score. In best-first search, we prioritize the states by the sum of log-likelihoods of tactics leading to that state.

## C.2   The GPT-4 Baseline

Now we describe the GPT-4 [27] baseline in Sec. 6. Similar to ReProver, it is a tactic generator combined with best-first search. However, the tactic generator is based on GPT-4's capability to follow instructions in zero-shot. Specifically, given a proof state, we use the following prompt to instruct GPT-4 to produce a list of tactics, each paired with a confidence score:

---

**Prompt Template:**
You are an expert in Lean3 theorem proofs. We are trying to solve the Lean3 theorem `THEOREM_FULL_NAME` from the mathlib file `FILE_PATH`. The current tactic state is: `TACTIC_STATE`. Suggest exactly 35 unique tactics to progress in solving `THEOREM_FULL_NAME`, along with their confidence levels as a float between 0 and 1. Rank them in order of effectiveness. Present the tactics and their confidence levels as comma-separated tuples in this format: #(tactic_{1}, confidence_{1})#, #(tactic_{2}, confidence_{2})#, ..., #(tactic_{35}, confidence_{35})#.

---

We adapted the prompt to a particular theorem and state by substituting the variables with the appropriate values. Given the inherent variability in GPT-4's outputs, we requested 35 and filtered out invalid ones. We used a token length limit of 1,024 and kept all other API parameters at their default values. Below are a few example prompts and GPT-4's responses:

---

**Example Prompt 1:**
You are an expert in Lean3 theorem proofs. We are trying to solve the Lean3 theorem `polynomial.chebyshev.aeval_U` from the mathlib file `mathlib/src/analysis/special_functions/trigonometric/chebyshev.lean`. The current tactic state is: `R : Type u_1, A : Type u_2, _inst_1 : comm_ring R, _inst_2 : comm_ring A, _inst_3 : algebra R A, x : A, n : ℕ ⊢ ⇑(aeval x) (chebyshev.U R n) = eval x (chebyshev.U A n)`. Suggest exactly 35 unique tactics to progress in solving `polynomial.chebyshev.aeval_U`, along with their confidence levels as a float between 0 and 1. Rank them in order of effectiveness. Present the tactics and their confidence levels as comma-separated tuples in this format: #(tactic_{1}, confidence_{1})#, #(tactic_{2}, confidence_{2})#, ..., #(tactic_{35}, confidence_{35})#.

---

**Example Response 1:**
#(rw chebyshev.U, 0.9)#, #(simp only [chebyshev.U], 0.9)#, #(simp_rw
chebyshev.U, 0.8)#, #(apply aeval_add, 0.75)#, #(apply aeval_mul,
0.75)#, #(apply aeval_sub, 0.75)#, #(apply aeval_X, 0.7)#, #(ap-
ply aeval_C, 0.7)#, #(apply aeval_pow, 0.7)#, #(rw \u2190aeval_pow,
0.65)#, #(rw Ž190aeval_mul, 0.65)#, #(rw \u2190aeval_add, 0.65)#, #(rw
\u2190aeval_sub, 0.65)#, #(apply algebra_map_eq, 0.6)#, #(intro ae-
val_algebra_map_eq, 0.6)#, #(apply eval_map, 0.55)#, #(rw eval_map,
0.55)#, #(apply aeval_eq_eval, 0.55)#, #(rw_for_target aeval_eq_eval,
0.5)#, #(apply aeval_eq_eval', 0.5)#, #(rw_for_target aeval_eq_eval',
0.5)#, #(apply aeval_commutes, 0.45)#, #(rw aeval_commutes, 0.45)#, #(ap-
ply aeval_eq_of_is_scalar_tower, 0.4)#, #(rw is_scalar_tower.aeval_apply,
0.4)#, #(try_for 100 exact_mod_cast chebyshev.U_eq_refl, 0.35)#, #(simp
* at *, 0.3)#, #(simp int.coe_nat_ring_hom, 0.25)#, #(apply cheby-
shev.U_is_polynomial, 0.2)#, #(rw_algebra_map_eq_of_ring_hom, 0.15)#,
#(unfold chebyshev.U, 0.1)#, #(congr, 0.05)#, #(change (chebyshev.U R
n).eval x = chebyshev.U A n, 0.05)#, #(change (\u21d1(aeval x) (cheby-
shev.U R n) = chebyshev.U A n), 0.05)#, #(apply_fun aeval x, 0.05)#,
#(apply_fun eval x, 0.05)#, #(simp context_md := reducible [], 0.01)#

**Example Prompt 2:**
You are an expert in Lean3 theorem proofs.  We are trying to solve
the Lean3 theorem 'euler_sine.integral_cos_pow_pos' from the math-
lib file 'mathlib/src/analysis/special_functions/trigonometric
/euler_sine_prod.lean'.  The current tactic state is:  'n :  ℕ ⊢ 0 <
∫ (x :  ℝ) in 0..π / 2, cos x ^ n'.  Suggest exactly 35 unique tactics
to progress in solving 'euler_sine.integral_cos_pow_pos', along with
their confidence levels as a float between 0 and 1.  Rank them in or-
der of effectiveness.  Present the tactics and their confidence levels
as comma-separated tuples in this format:  #(tactic_1, confidence_1)#,
#(tactic_2, confidence_2)#, ..., #(tactic_35, confidence_35)#.

**Example Response 2:**
#(exact integral_pos_of_antimono_increasing_on cos.pow_antimono_ in-
creasing_on_cos_even_nnreal, 0.99)#, #(apply integral_pos_of_antimono_
increasing_on, 0.95)#, #(interval_integrable_integral_pos, 0.90)#, #(ex-
act integral_pos_pre, 0.85)#, #(interval_integ_timedout, 0.80)#, #(in-
terval_integrable_pos, 0.75)#, #(exact integral_pos_of_nonneg_dint_val,
0.70)#, #(exact interval_integrable_pos_left, 0.65)#, #(exact inter-
val_ integrable_pos_right, 0.60)#, #(exact integral_pos_of_nonneg_f,
0.55)#, #(rw integral_on_interval, 0.50)#, #(rw interval_integral_pos_
of_supporting, 0.45)#, #(rw integral_pos_of_interval_integrable_pos,
0.40)#, #(apply integral_ pos_of_antimono_increasing_on_cos_even_nnreal,
0.35)#, #(norm_cast apply_pos, 0.30)#, #(apply_pos_of_nonneg_pos,
0.25)#, #(rw interval_integral_pos_of_antimono, 0.20)#, #(use
cos_slow_mono_on_pos, 0.15)#, #(apply cos_slow_mono_on_pos, 0.10)#, #(ex-
act integral_pos_pre_cos_pow_even, 0.09)#, #(norm_cast apply_nonneg_pos,
0.08)#, #(rw integral_pos_of_ monotone_integrable, 0.07)#, #(use
cos_pos_constant_sequence, 0.06)#, #(apply cos_pos_constant_sequence,
0.05)#, #(use integral_pos_of_ascending_sequence, 0.04)#,
#(apply integral_pos_of_ascending_sequence, 0.03)#, #(inter-
val_integral_pos_of_antimono_increasing_on_pos, 0.02)#, #(norm_cast
apply_pos_of_nonzero_pos, 0.01)#, #(apply_pos_of_nonzero_pos, 0.005)#,
#(rw integral_pos_of_nonneg_f_monotone, 0.001)#

Data contamination is possible. Our GPT-4 experiments were performed in 2023, but many theorems and proofs in the dataset have been publicly available on GitHub before GPT-4's data cutoff date (September 2021).

## C.3   Justifications for Not Comparing with Existing LLM-Based Provers

In Table 2, we do not empirically compare ReProver with any existing LLM-based prover. Unfortunately, such a comparison is infeasible. Provers targeting different proof assistants are generally not comparable, so we focus the discussion on the three existing provers in Lean [16, 17, 19]. Most importantly, they are impossible to reproduce with reasonable effort, due to private code and pretraining data. Therefore, the only potential comparison is to evaluate ReProver under their experimental settings and compare with the numbers reported in their papers. However, that is also impractical for numerous reasons:

- The data is different. All existing methods used an outdated version of `mathlib` more than two years ago. We cannot use LeanDojo to extract data from this version. As mentioned in Sec. 4, LeanDojo only supports repos released after March 24, 2022. Also, we cannot use their dataset directly, since it does not contain premise information required by ReProver.

- Lample et al. [17] trained on a synthetic dataset named Equations, which is not publicly available.

- All existing methods co-train the tactic generator on auxiliary tasks from the PACT dataset [16]. Co-training increases the data/compute requirements by an order of magnitude, which cannot be afforded by us (or probably most academic labs). All existing methods were developed by researchers in the industry.

- Polu et al. [19] and Lample et al. [17] further finetuned their models on new proofs collected through online interaction with Lean, whereas our method is only trained on human-written proofs.

- The tool for interacting with Lean may impact the performance. Han et al. [16] and Polu et al. [19] used `lean-gym`, which has severe limitations (Appendix A.2). Lample et al. [17] developed their own private tool, which is not publicly available.

Most of these difficulties are due to the private nature of existing methods. By releasing our code and models, we take a major step in establishing accessible baselines for future work to build upon.

## C.4   Evaluation on MiniF2F and ProofNet

We evaluate our ReProver model on MiniF2F [28] and ProofNet [29] (Sec. 6) to test its capability in proving theorems outside its training data distribution. We use the same hyperparameters and evaluation setup as the previous experiments (Appendix C.1).

**MiniF2F.**   We use the commit 5271ddec788677c815cf818a06f368ef6498a106 of Meta's version of MiniF2F [17]. ReProver achieves a Pass@1 of 26.5% on the test set, which is competitive with state-of-the-art methods without reinforcement learning (25.9% in Polu et al. [19]). Moreover, ReProver can prove 33 theorems that currently do not have Lean proofs (examples in Fig. B). For the complete list of 33 new proofs, please see our pull request to MiniF2F.

There are caveats about quantitatively comparing ReProver with existing methods on MiniF2F. Many difficulties in Appendix C.3 still apply, e.g., different tools for interacting with Lean may impact the performance. Also, MiniF2F is a test-only dataset without training theorems, and existing methods focus on reinforcement learning (RL) to learn from proofs collected via online interaction with the proof assistant [17, 19]. In contrast, ReProver is trained via supervised learning on a static dataset, so we only compare with the non-RL baseline in existing methods (Polu et al. [19] achieves a Pass@1 of 25.9% without RL and 29.6% with RL). Furthermore, we do not compare with Lample et al. [17] due to differences in the evaluation metric. They use Pass@64, which requires running the prover on each theorem 64 times. We use Pass@1, and it already takes one day for a single evaluation on MiniF2F's test set. Therefore, evaluating Pass@64 would be too computationally expensive for the resources we have access to. Finally, MiniF2F is available in multiple proof assistants [18, 69, 70]. Results across different proof assistants are not comparable, so we only compare with existing work in Lean.

**ProofNet.**   We use the commit e8645aa830ce17c33a8b8482a8195f0f97d6a74a of ProofNet. ReProver can prove 48 out of 349 theorems, achieving a Pass@1 of 13.8%, which is the first reported

```
theorem mathd_numbertheory_237 :
  (∑ k in (finset.range 101), k) % 6 = 4 :=
begin
  rw [finset.sum_range_succ'],
  norm_num [finset.sum_range_succ],
end

theorem mathd_numbertheory_175 :
  (2^2010) % 10 = 4 :=
begin
  norm_num [pow_succ],
end

theorem mathd_numbertheory_293
  (n : ℕ)
  (h₀ : n ≤ 9)
  (h₁ : 11|20 * 100 + 10 * n + 7) :
  n = 5 :=
begin
  contrapose! h₁,
  norm_num [h₁],
  dec_trivial!,
end

theorem mathd_algebra_616
  (f g : ℝ → ℝ)
  (h₀ : ∀ x, f x = x^3 + 2 * x + 1)
  (h₁ : ∀ x, g x = x − 1) :
  f (g 1) = 1 :=
begin
  simp only [h₀, h₁, pow_one],
  ring,
end
```

Figure B: Examples of new proofs discovered by ReProver on MiniF2F [28].

theorem proving result on ProofNet. Moreover, 39 out of the 48 proved theorems do not have existing Lean proofs (examples in Fig. C), and 3 of them can only be proved with the help of premise retrieval (Fig. D). We have contributed the 39 new proofs to ProofNet, which helped them reveal and fix problems in the formalization of 7 theorems (details in our pull request).

## D  LeanDojo for Lean 4

Lean 3 and Lean 4 are two incompatible major versions of Lean,[12] and both are widely used. Lean 3 was the latest stable version until recently (June 2023). Also, Lean 3 and Lean 4 have separate versions of mathlib. The Lean/mathlib community has recently finished porting theorems and proofs from mathlib 3 to mathlib 4 [100]. Therefore, Lean 3 will gradually become deprecated, and future Lean projects will be using Lean 4. Therefore, it is important for LeanDojo to support Lean 4.

Since Lean 4 is relatively new, we are not aware of any existing work on learning-based theorem proving in Lean 4. Furthermore, no existing tool is available for extracting data from Lean 4. LeanDojo fills in this gap and fully supports Lean 4. Given any repo in Lean 4, LeanDojo can extract

---
[12]https://leanprover.github.io/lean4/doc/lean3changes.html

```
theorem exercise_2_3_2 {G : Type*} [group G] (a b : G) :
  ∃ g : G, b * a = g * a * b * g⁻¹ :=
begin
  exact ⟨b, by simp⟩,
end

theorem exercise_11_2_13 (a b : ℤ) :
  (of_int a : gaussian_int) | of_int b → a | b :=
begin
  contrapose,
  simp,
end

theorem exercise_1_1_17 {G : Type*} [group G] {x : G} {n : ℕ}
  (hxn: order_of x = n) :
  x⁻¹ = x ^ (n − 1 : ℤ) :=
begin
  rw zpow_sub_one,
  simp,
  rw [← hxn, pow_order_of_eq_one],
end

theorem exercise_3_1_22b {G : Type*} [group G] (I : Type*)
  (H : I → subgroup G) (hH : ∀ i : I, subgroup.normal (H i)) :
  subgroup.normal (⊓ (i : I), H i):=
begin
  rw infi,
  rw ←set.image_univ,
  rw Inf_image,
  simp [hH],
  haveI := λ i, (H i).normal,
  split,
  intros x hx g,
  rw subgroup.mem_infi at hx ⊢,
  intro i,
  apply (hH i).conj_mem _ (hx i),
end

theorem exercise_3_4_5a {G : Type*} [group G]
  (H : subgroup G) [is_solvable G] : is_solvable H :=
begin
  apply_instance,
end
```

Figure C: Examples of new proofs discovered by ReProver on ProofNet [29].

data, including file dependencies, ASTs, proof states, tactics, and premise information. In addition, it enables the model to interact with Lean 4 through tactics, in the same way as Lean 3 (Sec. 4).

Similar to constructing the Lean 3 version of LeanDojo Benchmark, we extract data from the commit 3ce43c18f614b76e161f911b75a3e1ef641620ff of mathlib4 released on October 21, 2023. The resulting dataset is named LeanDojo Benchmark 4. It is released under the CC BY 2.0 license and hosted on zenodo.org with DOI "10.5281/zenodo.8040109". LeanDojo Benchmark 4 consists of 102,514 theorems/proofs, 213,067 tactics, and 152,695 premises. We use 2,000 theorems for

```
theorem exercise_13_6_10 {K : Type*} [field K] [fintype Kˣ] :
  ∏ (x : Kˣ), x = -1 :=
begin
  exact finite_field.prod_univ_units_id_eq_neg_one,
end

theorem exercise_1_17
  (n : ℕ)
  (x y : euclidean_space ℝ (fin n)) -- R^n
  : ‖x + y‖^2 + ‖x - y‖^2 = 2*‖x‖^2 + 2*‖y‖^2 :=
begin
  rw [norm_add_sq_real, norm_sub_pow_two_real],
  ring,
end

theorem exercise_2_25 {K : Type*} [metric_space K] [compact_space K] :
  ∃ (B : set (set K)), set.countable B ∧ is_topological_basis B :=
begin
  rcases exists_countable_basis K with ⟨B, hBc, hB⟩,
  exact ⟨B, hBc, hB.2⟩,
end
```

Figure D: Three new proofs discovered by ReProver on ProofNet [29] that cannot be found by a baseline without premise retrieval. All of the three proofs rely on premises: "`finite_field.prod_univ_units_id_eq_neg_one`"
, "`norm_add_sq_real`", "`norm_sub_pow_two_real`", and "`exists_countable_basis`".

validation, 2,000 theorems for testing, and the rest for training. LeanDojo Benchmark 4 also has two different data splits: `random` and `novel_premises`.

We use LeanDojo Benchmark 4 to train and evaluate our method. The model architectures and experimental details are the same as those in Sec. 6. Results on premise selection are in Table B, and results on theorem proving are in Table C.

Table B: Premise selection testing performance on LeanDojo Benchmark 4 (Lean 3 results in Table 1). We train and evaluate two models independently using different data splits (`random` and `novel_premises`). R@k is the recall for the top $k$ retrieved premises, and MRR is the mean reciprocal rank metric.

| Method | random | | | novel_premises | | |
|--------|------|-------|------|------|-------|------|
| | R@1 | R@10 | MRR | R@1 | R@10 | MRR |
| Ours | 12.8 | 34.7 | 0.29 | 9.8 | 32.1 | 0.24 |

Table C: Theorem proving Pass@1 (%) on the testing data of LeanDojo Benchmark 4 (Lean 3 results in Table 2).

| Method | random | novel_premises |
|--------|--------|----------------|
| ReProver | **48.6** | **19.9** |
| W/o retrieval | 44.5 | 16.2 |

# E   ChatGPT Plugin for Theorem Proving

LeanDojo provides a general tool for interacting with Lean programmatically. As a demo of how it might bridge LLMs and theorem proving, we build a ChatGPT plugin [101] enabling ChatGPT to prove theorems by interacting with Lean through LeanDojo. Plugin developers can wrap any software

as a web service and describe its APIs to ChatGPT. Then, ChatGPT can automatically call the APIs and incorporate the results into the response to the user. Below is a summary of our API description corresponding to the interface in Sec. 4.

```
Title: Lean

Description: Plugin for proving user-specified theorems
    automatically by interacting with Lean. The user enters
    information of how to find a theorem (e.g., theorem name
    and file path). Based on the user's input, ChatGPT first
    initializes the proof search with the given theorem as the
    initial state. Then, ChatGPT will first explain the choice
    for the next tactic step using LaTeX and run that tactic
    step to the state. If the current state is not promising,
    ChatGPT can backtrack to previous states by decrementing
    the "state_id" parameter. If applying tactics to the
    current state specified by the "state_id" parameter returns
     an error message, ChatGPT should explain the error, and if
     repetitive errors occur, ChatGPT should decrement the "
    state_id" parameter and try a different approach on a
    previous state. The theorem is successfully proved if there
     are no unsolved goals in the current state.

Endpoints:
    initialize_proof_search: Given the theorem name and file
        path of a Lean theorem, initialize the proof search.
        The response includes the initial state and its state
        ID.
    Args:
        theorem_name (string): The name of the target theorem
            to prove.
        theorem_file_path (string): The file path of the target
             theorem.

    run_tactic: Run a tactic on a state (specified by its state
        ID), assuming the proof search has been initialized
        and some state is available. The response is either the
         next state and its state ID or an error message, in
        which ChatGPT should explain the error and consider
        decrementing the "state_id".
    Args:
        state_id (string): The ID of the state on which to run
            the tactic.
        tactic (string): The tactic to run on a state (
            specified by its state ID), assuming the proof
            search has been initialized.
```

After exposing the APIs to ChatGPT, we can ask it to prove theorems by specifying the theorem's name and path in any public Lean repo on GitHub. Fig. E–L show an example with the GPT-3.5 version of ChatGPT. And Fig. M–O are the same example with the GPT-4 version. The captions provide detailed step-by-step explanations.

We highlight a few key strengths of ChatGPT observed in multiple examples we evaluated. First, unlike specialized methods for theorem proving (this paper and its prior works), ChatGPT interleaved informal mathematics with formal proof steps. This resembles how humans interact with proof assistants and opens up new avenues for integrating natural language and formal theorem proving. Second, ChatGPT demonstrated impressive capability in explaining error messages from Lean that are quite opaque even to humans. It was able to incorporate the error message to refine its proof strategy. Last, ChatGPT's behavior is more steerable than specialized provers. In Fig. E, we simply gave it the theorem to prove, but we could also provide more detailed instructions. For example, we

could say: "Please describe a high-level proof plan before trying any tactic." This kind of steerability enables future research on prompt engineering for theorem proving, and we have already seen initial benefits in an ongoing work named Sagredo.[13]

However, these strengths by no means imply ChatGPT can already solve theorem proving. In fact, it failed to find a proof for most theorems we tried. Hallucination was common. In Fig. L, ChatGPT falsely asserted the theorem was proved, while we knew it was not, by looking at LeanDojo's response. This demonstrates the value of theorem proving as a rigorous benchmark for addressing LLMs' hallucination problem. Another key limitation of ChatGPT was its inability to search systematically in a large space. We frequently found it stuck to an unpromising path when the correct solution could be found by backtracking and exploring alternative paths. This behavior is consistent with the general observation that LLMs are weak at search and planning. Addressing this weakness is an active area of research [102].

We emphasize a few caveats about our study of theorem proving with ChatGPT. First, data contamination is likely. Many theorems we evaluated have been publicly available on GitHub before ChatGPT's data cutoff date. Therefore, ChatGPT may have seen them in training. Second, our study is exploratory. A more detailed and quantitative study is needed to characterize ChatGPT's capability in theorem proving. Such a study with ChatGPT plugins is challenging, as plugins currently only support interaction through the browser. Also, OpenAI has taken measures to block automated access by bots. Using humans may be an option, but that is beyond the scope of this paper.

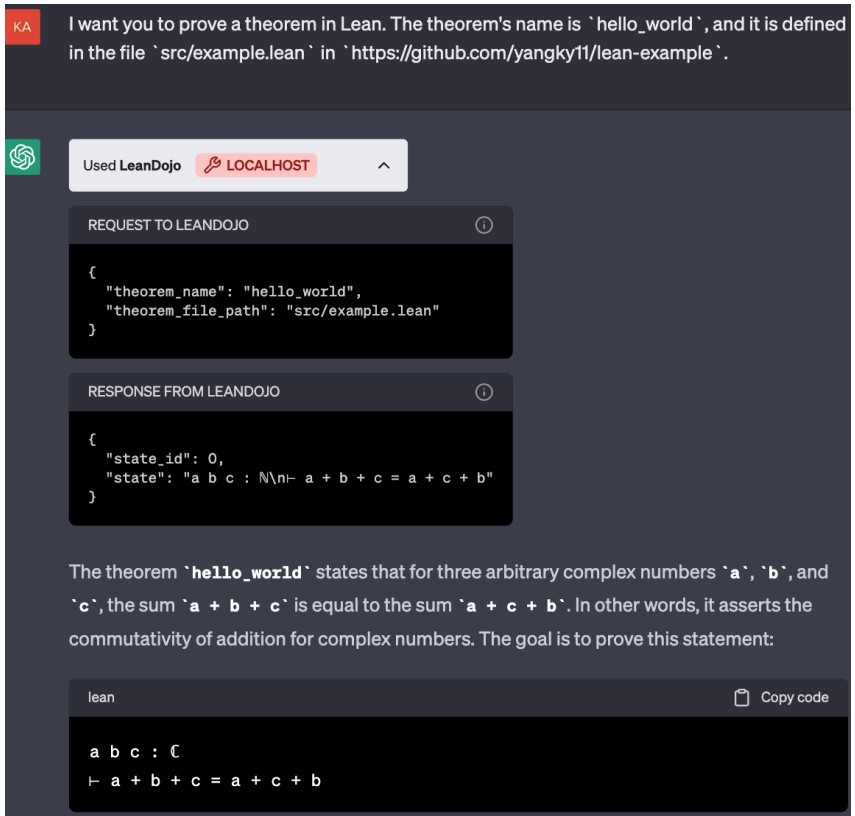

Figure E: (ChatGPT-3.5, 1/8) After receiving the theorem to prove, ChatGPT first called "`initialize`", which returned the initial state. Then it tried to interpret the theorem in natural language. Note that it made a mistake here. The theorem was about natural numbers ($\mathbb{N}$), not complex numbers ($\mathbb{C}$).

---

[13] https://www.youtube.com/watch?v=CEwRMTOGpKo

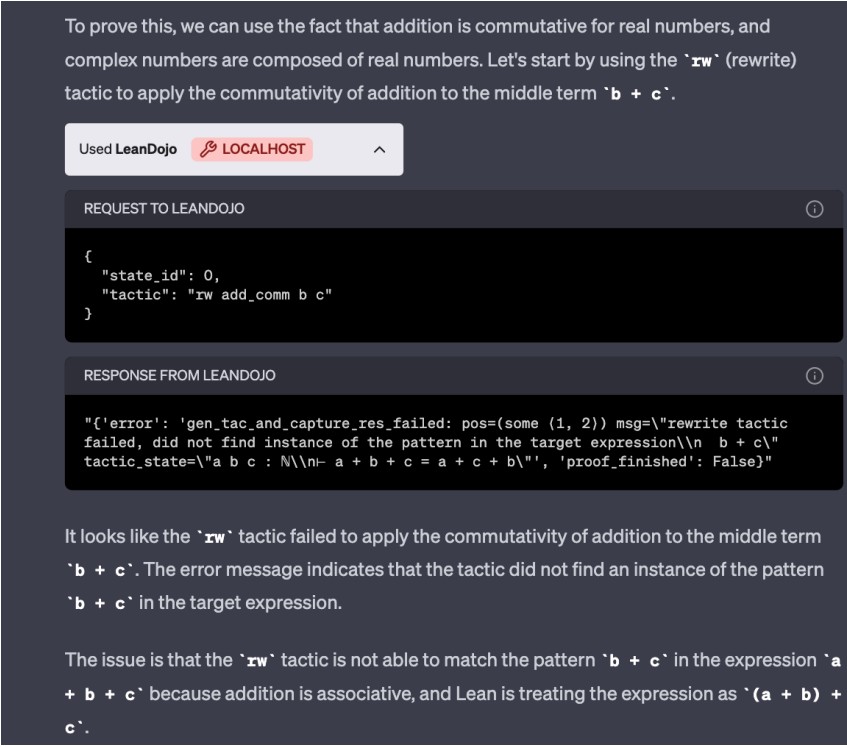

Figure F: (ChatGPT-3.5, 2/8) ChatGPT tried to rewrite the goal using the lemma "b + c = c + b". This was a reasonable but incorrect move. After receiving the error message from Lean, ChatGPT explained the error in natural language. Here the explanation is quite accurate, which is impressive given that the error message looks opaque to anyone not familiar with Lean.

## F    Limitations and Future Work

Our work is one step toward unlocking the potential of LLMs for generating verifiable formal proofs, and we see abundant space for future exploration. A learning-based prover is a complex system consisting of multiple components: data extraction, interaction with proof assistants, model training, and proof search. While navigating the design space spanned by various components, we err on the side of simplicity and efficiency, instead of pushing performance to the limit. This helps us deliver a reliable, open, and accessible system, laying the foundation for further research. There are many directions in which the system can be improved, and we discuss a few of them here.[14]

**Stronger LLMs.**    Our backbone model, ByT5 [44], was published in 2021 and has 299M parameters, which is not very large by today's standard. Recently, there have been a plethora of open-source LLMs demonstrating strong capabilities in writing code, e.g., CodeGen [103], StarCoder [94], and CodeGeeX [104]. We are excited to see how they might impact theorem proving and, more generally, how far we can go by pushing the limit of the model/data scale.

ByT5's tokenizer-free nature helps us sidestep the difficulty with pretrained tokenizers that may not work well for Lean's Unicode-rich code. However, treating texts as raw bytes makes the sequence length much longer than necessary. Long sequences harm efficiency, as Transformers scale quadratically w.r.t. the sequence length, which may become a bigger problem when we further scale up the model. To solve the issue, it might be helpful to pretrain a customized tokenizer or adopt more advanced tokenizer-free models such as MegaByte [105].

Our ReProver model is based on the pretraining-finetuning paradigm. Recent work on instruction-following LLMs such as GPT-4 [27] has led to successes in many applications by prompting the model without any finetuning. Our preliminary results show that GPT-4 and ChatGPT (Appendix C.2

---

[14]Additional limitations: `https://leandojo.readthedocs.io/en/latest/limitations.html`

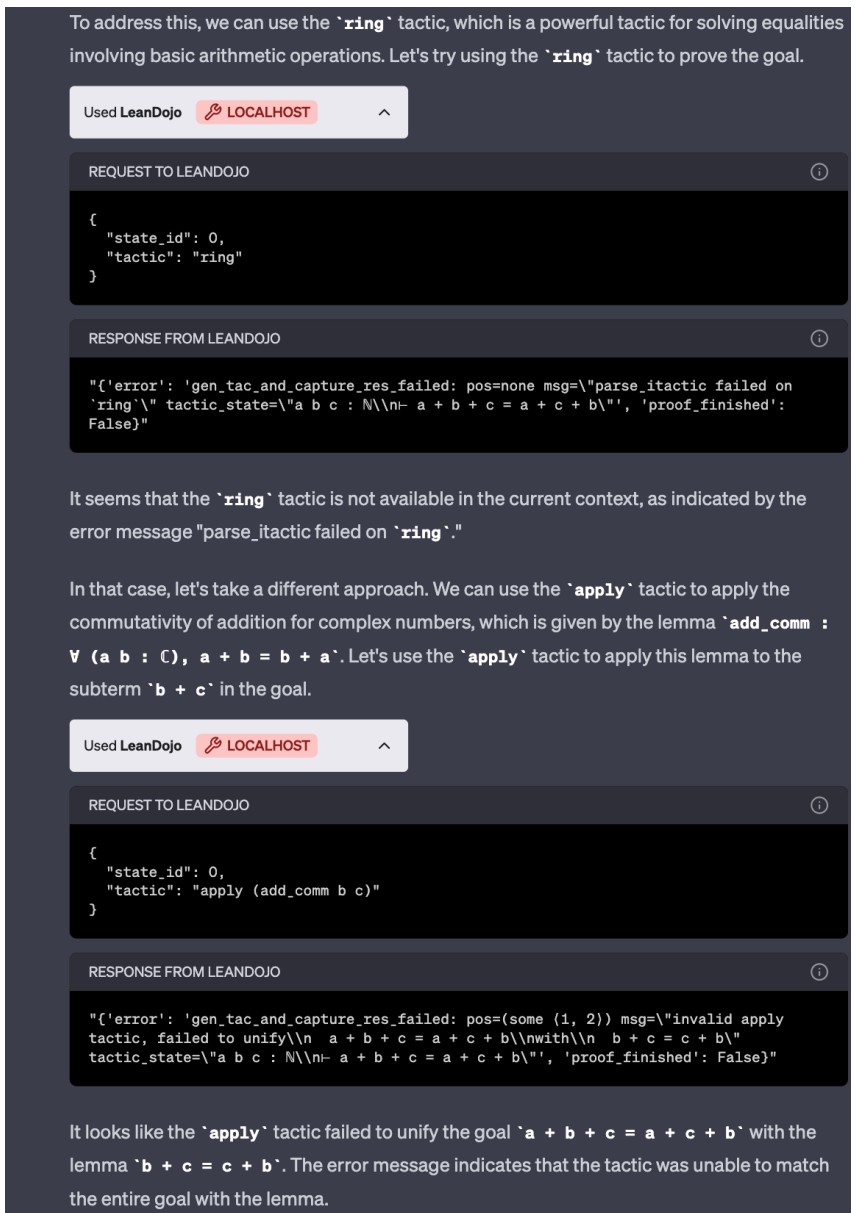

Figure G: (ChatGPT-3.5, 3/8) Then it tried to prove the theorem using "`ring`". This was another good move. The `ring` tactic can prove this theorem, but Lean could not find it since it was not imported into the current file. Again, ChatGPT was able to interpret the error message correctly and concluded that `ring` was not available. Next, it tried another tactic but failed again.

and E) cannot solve theorem proving out of the box and are currently far behind finetuned models. However, the way we prompt these models is quite naive, and better strategies, such as Tree of Thoughts [102], may lead to further improvements. We consider theorem proving as a promising task for studying LLMs' capabilities in planning and search.

**Improving Premise Retrieval.**    ReProver uses DPR [26] to retrieve premises and fuses them with the current proof state by concatenation. This architecture is simple and effective but does not scale to a large number of retrieved premises. With a length limit of 2,300 tokens, we can fit only 10–15 premises into the input of the tactic generator. To mitigate the problem, we may need an architecture that fuses the retrieved premises in the hidden space, e.g., Fusion-in-Decoder [106].

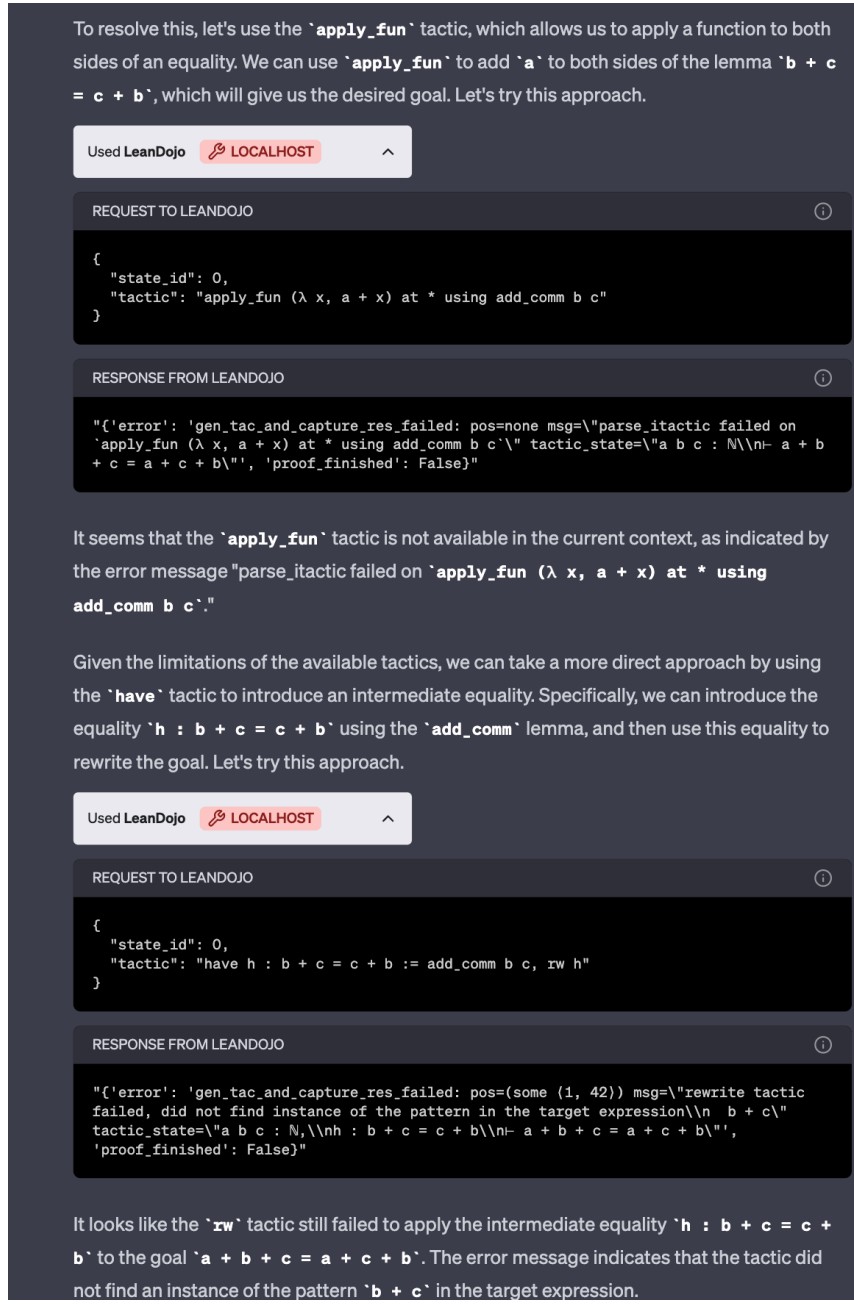

Figure H: (ChatGPT-3.5, 4/8) ChatGPT made another two failed attempts. Here, the second attempt had the same problem as in Fig. E ("+" is left associative).

In addition, one can also switch from DPR to radically different retrieval architectures. For example, generative retrieval [107–109] is a recent class of models performing retrieval by directly predicting the document IDs, which could be the premise names in our task.

**Limitations of Imitating Human-Written Proofs.** Human-written proofs extracted by LeanDojo provide valuable data for training the prover. However, we have also observed limitations of using them as the sole training target:

First, they are relatively scarce for today's data-hungry LLMs. LeanDojo Benchmark has 98,734 proofs, covering a large portion of available data in Lean (as of October 2023). The number of proofs

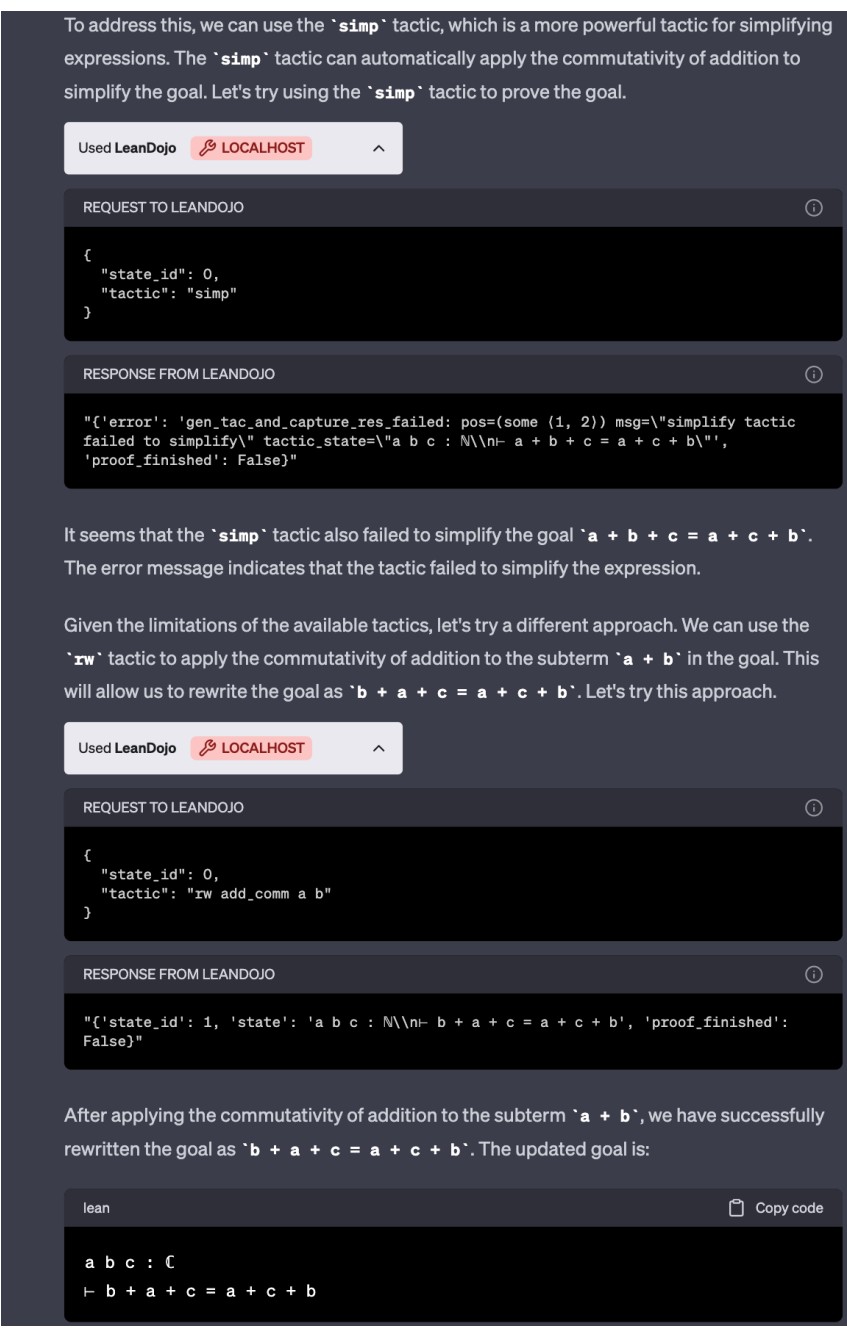

Figure I: (ChatGPT-3.5, 5/8) ChatGPT tried to simplify the goal using the simp tactic but failed. Then it tried the "rw add_comm a b" tactic, which successfully rewrote the goal from "a + b + c = a + c + b" to "b + a + c = a + c + b".

in other proof assistants has the same order of magnitude (tens or hundreds of thousands). Due to limited data, we cannot constantly improve the performance simply by scaling up the model size. Second, theorem proving in proof assistants is an interactive process, but the proof only captures the final successful trajectory. Without the intermediate history of trial and error, it can be quite opaque how final proofs are derived. Therefore, tactics in human-written proofs can be difficult for the model to learn from. Third, models trained on proofs in one project often struggle to generalize to theorems in new domains [9, 20], e.g., from mathlib to MiniF2F and ProofNet (Appendix C.4).

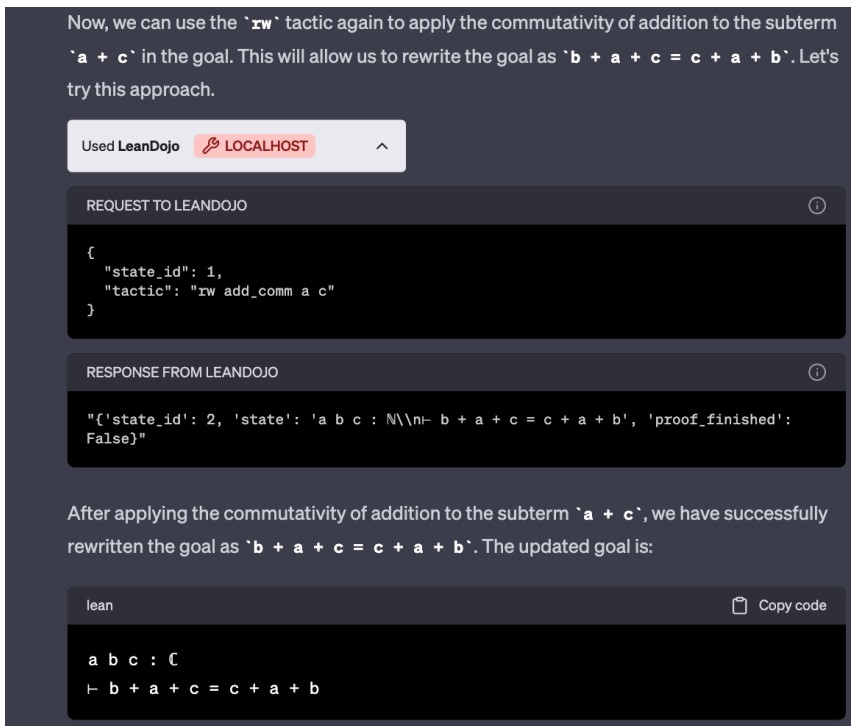

Figure J: (ChatGPT-3.5, 6/8) Then it successfully performed another rewrite from "b + a + c = a + c + b" to "b + a + c = c + a + b".

To overcome these limitations, existing work has explored learning from auxiliary data or data collected via online interaction with the proof assistant. For example, Proof Artifact Co-Training (PACT) co-trains the tactic generator on nine auxiliary tasks, such as predicting types and theorem names [16]. MetaGen [38] trains a neural network to generate synthetic theorems/proofs as training data in the Metamath proof assistant [110]. Polu et al. [19] and Lample et al. [17] improve the prover by training it on successful proofs found by itself. Incorporating these techniques into our system may lead to substantial improvements.

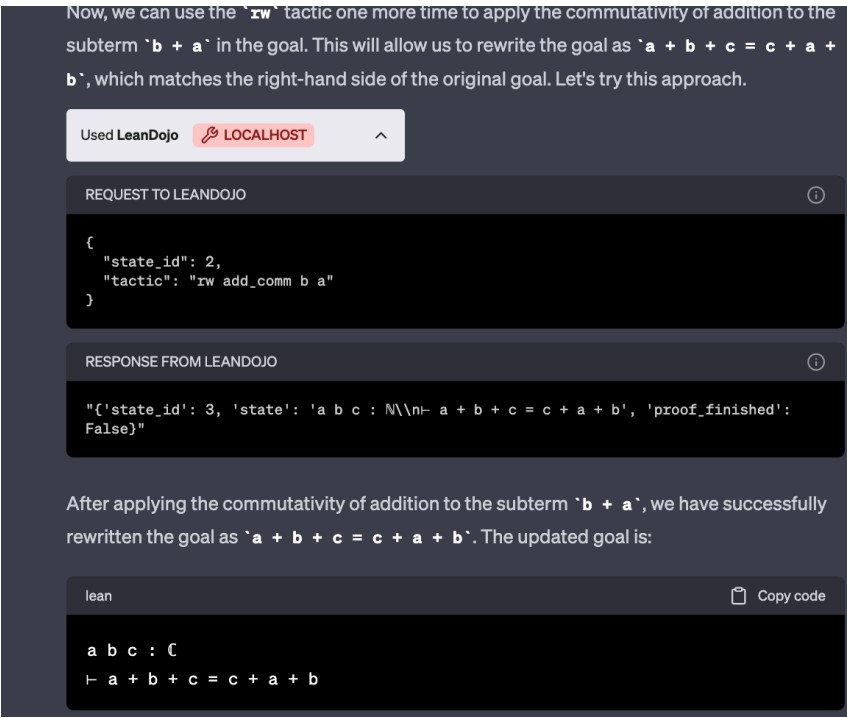

Figure K: (ChatGPT-3.5, 7/8) Another rewrite tactic, transforming the goal to "a + b + c = c + a + b".

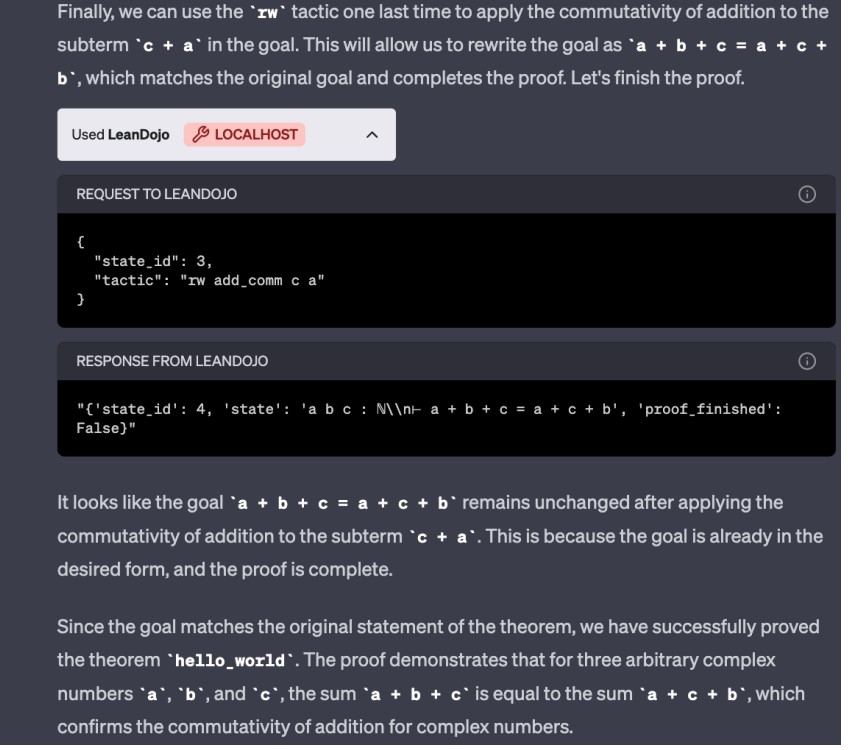

Figure L: (ChatGPT-3.5, 8/8) It performed one last rewrite, which brought the goal back to the original goal "a + b + c = a + c + b". At this point, ChatGPT went back to the starting point without making meaningful progress. However, it asserted that the theorem had been successfully proved, even though the response from LeanDojo clearly indicated "proof_finished:   False".

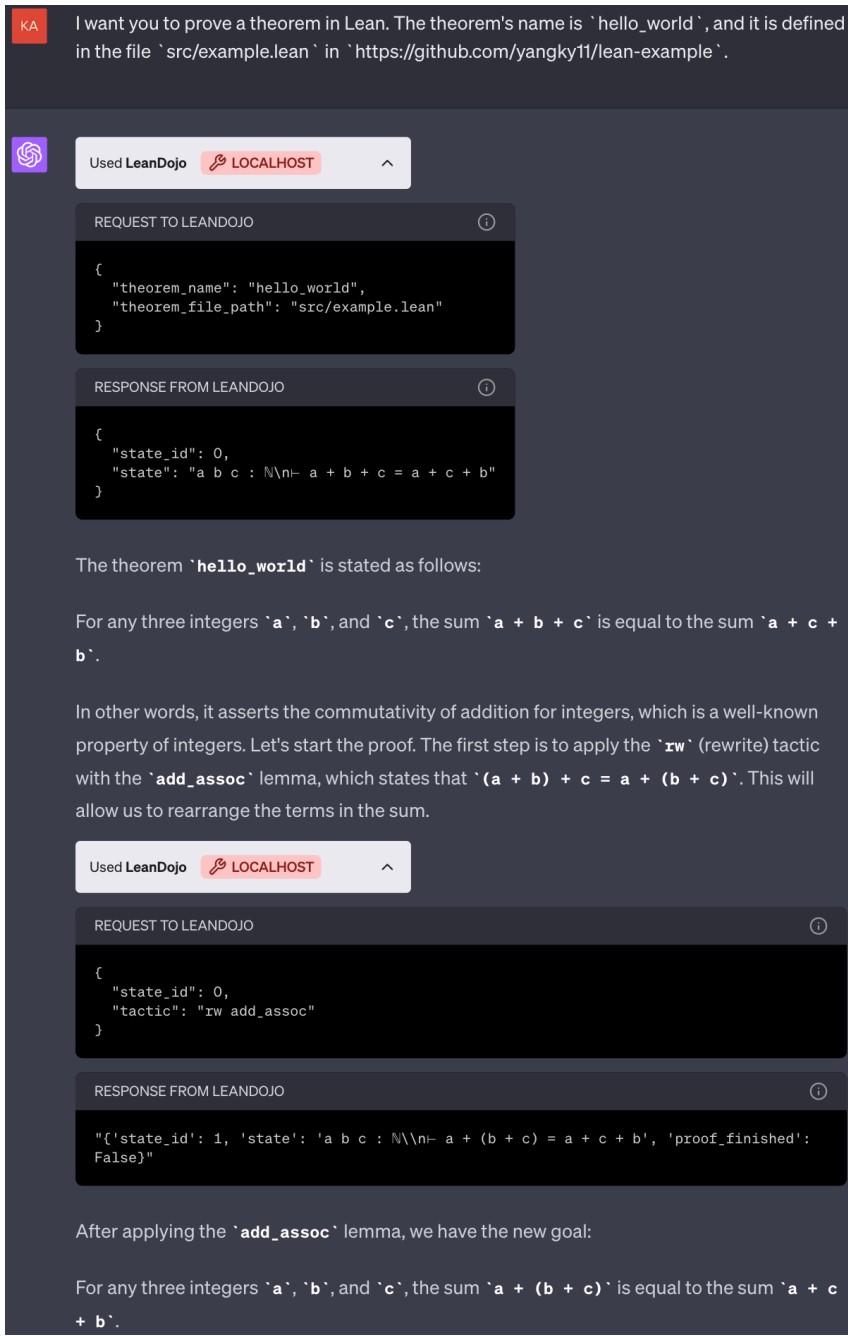

Figure M: (ChatGPT-4, 1/3) The same theorem using the GPT-4 version of ChatGPT. Despite a minor error when interpreting the theorem (natural numbers, not integers), it generated a sensible tactic, rewriting the goal to "a + (b + c) = a + c + b".

The next step is to apply the `rw` (rewrite) tactic with the `add_comm` lemma, which states that `b + c = c + b`. This will allow us to swap the terms `b` and `c` in the sum.

Used **LeanDojo**    🔧 LOCALHOST         ⌃

REQUEST TO LEANDOJO                                              ⓘ

```
{
  "state_id": 1,
  "tactic": "rw add_comm b c"
}
```

RESPONSE FROM LEANDOJO                                           ⓘ

```
"{'state_id': 2, 'state': 'a b c : ℕ\\n⊢ a + (c + b) = a + c + b', 'proof_finished':
False}"
```

After applying the `add_comm` lemma, we have the new goal:

For any three integers `a`, `b`, and `c`, the sum `a + (c + b)` is equal to the sum `a + c + b`.

Figure N: (ChatGPT-4, 2/3) A good move, rewriting the goal to "a + (c + b) = a + c + b".

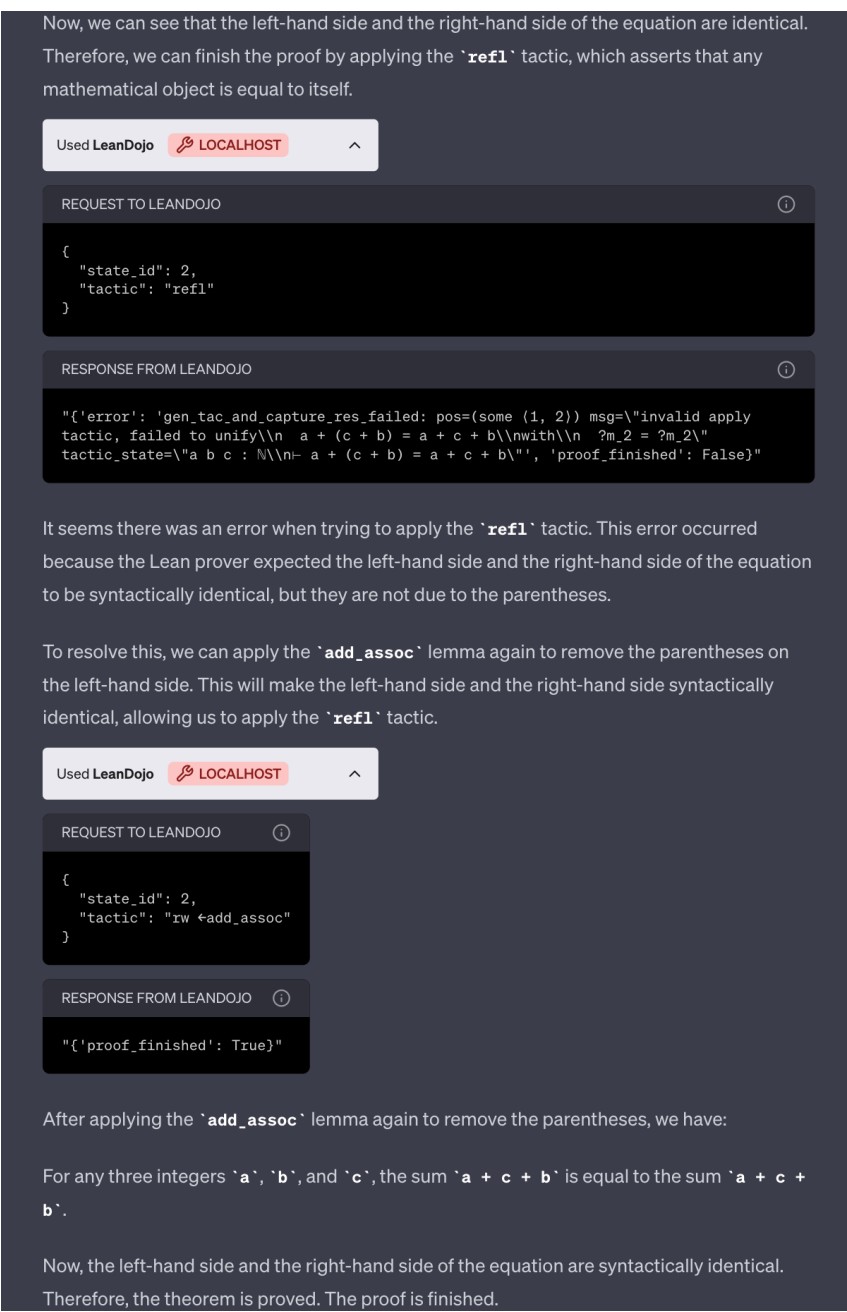

Figure O: (ChatGPT-4, 3/3) It tried the `refl` tactic but failed. `refl` requires the goal to be an equation whose both sides are identical up to trivial transformations. However, "a + (c + b) = a + c + b" is not trivial since "+" is left associative. ChatGPT was able to interpret this error accurately and finish the proof using the correct premise "`add_assoc`".

