# OpenReview forum: "LeanDojo: Theorem Proving with Retrieval-Augmented Language Models"
_NeurIPS.cc/2023/Track/Datasets_and_Benchmarks — NeurIPS 2023 Datasets and Benchmarks Oral_

### Official Review · Reviewer_fFD9 · 2023-06-29
**Review of the paper "LeanDojo: Theorem Proving with Retrieval-Augmented Language Models"**

**Rating:** 7
**Confidence:** 3
**Clarity:** Yes.

**Strengths:**

Lean is one of the most significant theorem provers and its size is very appropriate for machine learning. Given that the community is very actively "translating" mathlib to Lean 4, the work here might be of interest to many members of the community.

The paper is well structured and does not go into too many details regarding Lean, so it is accessible to the machine learning community. All the components of LeanDojo are well described, and the paper is easy to follow.

The experimental setup is mostly well-designed and the results are encouraging.

**Additional Feedback:**

1. It seems that almost all the materials (excluding both pdfs (the paper itself and the supplementary materials)) were put online after the deadline - this includes the webpage - which was "under construction" even in the bidding period (https://leandojo.org/), the dataset (e.g., https://zenodo.org/record/8040110) and the relevant github repositories (https://github.com/lean-dojo). Were they accessible anywhere before the deadline? (The current Rating of the paper assumes they were.)

2. Line 205: The paper says "In addition, LeanDojo produces the AST of each file." Inspecting the dataset, I was not able to find any ASTs. The dataset (e.g., leandojo_benchmark_4/random/test.json) contains files, imports and source code instead of the ASTs - as shown in the supplementary materials. Are ASTs available as well?



**Correctness:**

The claims made in the submission are correct, except for the one mentioned above: I don't agree that 120 hours of GPU training is a low number.

The experimental design is sound.

**Documentation:**

Yes.

**Ethics:**

No.

**Limitations:**

There is probably no potential negative societal impact of their work.

The limitations of the method has been briefly mentioned, but not explained (in contrast to the limitaions of the competing methods).
For example, the authors explain why lean-gym fails to type-check 21% of the proofs, but do not explain why Lean Dojo failes to proof-check 1.4% of them.

**Opportunities For Improvement:**

Even though the data-extraction tools is supposed to extract ASTs of the files as well, I was not able to find them in the dataset webpage(s). In the, for example, test.json file, only the source code is present, which might be hard to "parse" by a machine learning methods.

The in-file negative examples seem to help. However, this is based on human observation of large quantities of data and I wonder whether training a model in two (or more) iterations would be better:
- first randomly choose training examples (as the authors did originally)
- using internal cross-validation on the training set, create predictions: a lot of "false positives" should appear and they can be used instead of "in-file" negatives in the next iteration of learning.

I do not fully agree with the claim that the training of ReProver is efficient. In comparison to other models maybe, but in absolute numbers, 120 hours of GPU time is quite a bit and not accessible to everybody.


**Relation To Prior Work:**

Yes.

**Summary And Contributions:**

The paper describes a tool for extracting machine-learnable data from Lean, an extracted data set that was used in the experiments (extracted from Lean 3), and a model (ReProver) for automatic theorem proving - LeanDojo for short.

The data extraction tool is written in Lean and wrapped with Python.
The dataset comes with a predefined train/validation/test split, and is a directed-acyclic graph with mathlib files as nodes, and directed edges that resemble imports. Every file is represented as an abstract syntax tree (AST).

ReProver is claimed to be efficient and is based on a transformer that encodes pairs (state, premise) into vectors. It is trained to minimize `(l_ij - cos(state, premise))^2`, where `l_ij` is either 1 (positive example) or 0 (negative example).

---

> ### Author Response · Authors · 2023-08-20
> **Response to Reviewer fFD9**
>
> Dear reviewer,
>
> Thank you for your positive evaluation and constructive feedback! Below we address your questions and concerns. Please feel free to follow up if you have further questions!
>
>
> ## Where Are the Extracted ASTs?
>
>
> LeanDojo can extract ASTs, though they were not included in the released datasets due to postprocessing. LeanDojo’s documentation has a [Getting Started](https://leandojo.readthedocs.io/en/latest/getting-started.html) page providing simple examples of extracting ASTs in XML format. The user can easily extract the ASTs corresponding to our released datasets by specifying the right GitHub URL and commit hash (e.g., https://github.com/leanprover-community/mathlib4, `355541ae7a2455222f179dcf7f074aa2c45eb8aa`).
>
>
> ## Hard Negative Mining
>
>
> If we understand correctly, the reviewer proposes hard negative mining using the trained retriever. We experimented with this strategy earlier but didn't observe any performance improvements. One challenge was that model-selected hard negative premises were often wrong: although the ground truth proof didn't use them, they could be used to prove the theorem. Therefore, they were more like positive examples instead of hard negatives. Random negatives and in-file negatives suffer less from this issue.
>
>
>
> ## Computational Cost
>
> We agree a training time of 120 GPU hours may not be accessible to everyone. Nevertheless, we'd like to clarify a few points. First, the training can be run on a single GPU. Second, the 120 hours consist of two parts: training the premise retrieval + training the tactic generator. If you want to use either of them alone, its training takes only ~60 GPU hours. Third, since we open-source the model checkpoints, other researchers can finetune them on their own datasets with low computational costs.
>
>
> ## More Detailed Limitations. Why Does LeanDojo Fail on 1.4% of Proofs?
>
>
> LeanDojo’s documentation contains a dedicated page for [technical limitations](https://leandojo.readthedocs.io/en/latest/limitations.html). We have updated the paper to include a link to this page. Regarding the 1.4% failed proofs, we have sorted the errors into a few categories and documented a few representative examples of each category in [LeanDojo's unit tests](https://github.com/lean-dojo/LeanDojo/blob/main/tests/interaction/test_unexpected_errors.py). The causes of these errors remain unknown. We consulted core developers of Lean. They believe the errors are due to some intricate interplay between Lean 3’s implementation and how metaprogramming is used in LeanDojo. Furthermore, we have verified that our 1.4% failed proofs are a strict subset of lean-gym’s 21% failed proofs.
>
>
>
>
>
> ## Timeline for Releasing the Data, Code, and Website.
>
> The initial versions of the datasets were released on June 14 (Lean 3) and June 20 (Lean 4). The code was released on June 26, and the website was released on June 27. We didn't publicly release them earlier due to cleaning the code, adding documentation, and polishing the website.
>
> From our understanding of the [Call for Datasets & Benchmarks](https://neurips.cc/Conferences/2023/CallForDatasetsBenchmarks), it is not required to publicly release the data/code by the deadline. We apologize if their delayed release has caused inconvenience to the reviewing process.

---

### Official Review · Reviewer_xf1r · 2023-07-21
**Very interesting paper and good enough**

**Rating:** 8
**Confidence:** 5
**Correctness:** NO
**Clarity:** Well written

**Strengths:**

(1) the authors introduced LeanDojo, an open-source tool for theorem proving that fills the gap in existing resources. In addition, they proposed a retrieval-augmented prover named ReProver, addressing the critical issue of premise selection in theorem proving.
(2) they have significantly reduced computational requirements, making it possible to train the model within five days on a single GPU. This increases the model's practicality and accessibility.
(3) the paper is committed to the open-source ethos by providing the data, model, and code to the research community. This contribution will significantly facilitate further research in this field.






**Additional Feedback:**

NO.

**Documentation:**

Yes.

**Ethics:**

NO.

**Limitations:**

The content of this article is very comprehensive and substantial. It is a very interesting and excellent article.

**Opportunities For Improvement:**

As the authors mentioned in the supplementary material, I am also very curious about what would happen if the underlying language model is replaced with more powerful LLMs. Actually,  the author of this article has made significant contributions to the research in Theorem Proving. If possible, I am eagerly looking forward to this work playing a larger role in the field of mathematics in future research.

**Relation To Prior Work:**

Yes

**Summary And Contributions:**

The authors of this research paper introduce LeanDojo, which is an open-source Lean playground consisting of toolkits, data, models, and benchmarks. Then they develop ReProver (Retrieval-Augmented Prover), the first LLM-based prover that is augmented with retrieval for select ing premises from a vast math library. Finally, the authors construct a new benchmark consisting of theorems and proofs extracted from Lean’s math library. The experimental results demonstrate the effectiveness of ReProver over baselines and GPT-4.

The main contributions of this work are as follows:
 (1) tools for extracting data from and interacting with Lean.
 (2) the first retrieval-augmented language model (ReProver) for theorem proving.
 (3) benchmark for learning-based theorem proving and use it to validate the effectiveness of ReProver
 (4) open research on LLMs for theorem proving by releasing our data, model, and code.

---

> ### Author Response · Authors · 2023-08-20
> **Response to Reviewer xf1r**
>
> Dear reviewer,
>
> Thank you for your enthusiasm for our paper! We address your comments below. Please feel free to follow up if you have further questions!
>
>
>
>
> ## More Powerful LLMs
>
> We're actively working on replacing the ByT5 used in this paper with more powerful LLMs such as StarCoder, SantaCoder, and ProofGPT. Preliminary results have revealed a few challenges. First, Lean is quite unique among other programming languages in its extensive use of Unicode characters. We find off-the-shelf pretrained models may struggle with tokenizing Lean code due to Unicode. Second, most state-of-the-art LLMs are decoder-only models, whereas tactic generation can be handled more naturally by encoder-decoder models. We're investigating the performance implication of these challenges.
>
>
> ## Potential Impact on Formal Mathematics
>
> We are excited to explore the impact of neural theorem proving on formal mathematics! There are many exciting future directions, for example, we're trying to train machine learning models in Python, convert them to intermediate representations such as ONNX, and run the inference directly in Lean via Foreign Function Interface (FFI). If successful, it will deeply integrate Lean and machine learning. For example, it will enable people to build LLM-based tactic generators or premise retrievers directly in Lean, which work seamlessly in Lean’s VSCode workflow.

---

### Official Review · Reviewer_AsmJ · 2023-07-21
**Awesome open-source dataset and toolkit for formal theorem proving**

**Rating:** 9
**Confidence:** 4
**Clarity:** The paper is very well-structured and…

**Strengths:**

* This work contributes very valuable open resources to the theorem proving community. It is also very relevant to the broader AI community, since theorem proving is one of the most difficult open problems in AI.
* This work also has potential to make open-source contributions to the Lean interactive theorem prover. It is shown in the paper that the proposed model automatically found new proofs for dozens of theorems in the library that are still missing proofs.


**Additional Feedback:**

None

**Correctness:**

The construction of dataset and benchmark evaluation look correct. There are a few issues that I hope authors can clarify:
1. It is mentioned in Section 4 that there are two ways to split the dataset, namely “random” and “novel_premises”. However, I wasn’t able to find the discussion on which split is used to train the ReProver models. In Table 1&2, the authors report performance on the test set of both splits. Since each of the two splits seem to be a full partition of the entire dataset (Appendix B Line 73), if the model is trained on one split’s training set and evaluated on the other split’s test set, there might be data leakage issue. Hopefully the authors can clear this potential suspicion.
1. Line 291 says that in premise selection, the pool of candidate premises is first limited to those accessible from the current theorem, largely by following file import commands. I wonder if this may risk inflating the performance of premise selection, because when proving a new theorem, knowing which files contain useful premises might not be a safe assumption to make.


**Documentation:**

The dataset is documented in supplementary materials Section B. Dataset is hosted on Zenodo, code is hosted on GitHub, and models are hosted on HuggingFace.

**Ethics:**

There seems to be no ethical concern.

**Limitations:**

Limitations have been comprehensively discussed in the supplementary material. There seems to be no direct societal impact with this work.

**Opportunities For Improvement:**

The paper looks very solid overall. I made a few comments under the “Correctness” section.

**Relation To Prior Work:**

Related work is well-discussed. Minor comments:
1. The following claim may not be very accurate: “we are the first to augment a learning-based prover with retrieved premises so that the prover can learn how to use them effectively” (Line 132). For example, NaturalProver (Welleck et al., 2022) features a retrieval-augmented theorem prover that works with informal mathematical language.

**Summary And Contributions:**

This paper presents a suite of resources for the task of theorem proving in Lean using language models. This includes a dataset of Lean theorems and proofs, the associated tools for interacting with the Lean environment, and a strong model trained on this dataset. All resources are made publicly available to facilitate research in theorem proving.

---

> ### Author Response · Authors · 2023-08-20
> **Response to Reviewer AsmJ**
>
> Dear reviewer,
>
> Thank you for your positive evaluation and constructive feedback! Below we address your questions and concerns. Please feel free to follow up if you have further questions!
>
>
> ## Which Data Split Was Used to Train ReProver?
>
> Results in Table 1 and Table 2 were produced by two independently trained models. One model was trained/tested on the training/testing set of the “random” split, and the other model was trained/tested on the training/testing set of the “novel_premises” split. Thanks for pointing out the confusion, and we have updated the table captions to clarify.
>
>
>
> ## Having Access to All Imports May Inflate the Performance
>
> In the existing task setup (adopted by our work and prior works, e.g., [14–20]), the model is allowed to access all information before the proof (including file imports) and is only responsible for generating the proof. We agree with the reviewer that it is a simplification of the task faced by humans. When humans prove a theorem, they may need to import the correct premise by adding additional import statements. This can be an interesting alternative task setup, though we are not aware of prior works adopting this setup.
>
>
>
>
>
> ## Relation to NaturalProver (Welleck et al. 2022)
>
> Thank you for pointing out the confusion. We have updated the paper to clarify that “prover” here refers only to formal theorem provers.

---

> > ### Comment · Reviewer_AsmJ · 2023-08-27
> >
> > Thanks for your response! I think all my comments have been well addressed. Kudos to the authors for the awesome work!

---

### Official Review · Reviewer_crGr · 2023-07-22
**Useful Open-Source Benchmark**

**Rating:** 8
**Confidence:** 2
**Correctness:** Given my limited understanding of the…

**Strengths:**

- A new dataset for building stronger theorem provers
- A benchmark for fair and reproducible comparison of theorem provers
- An efficient retrieval-based approach that outperforms prior work.

**Additional Feedback:**

- Are the generated proofs for new tasks, by design, always correct? If not, did you verify what % of the new proofs are correct? If yes, can we use this system in a self-training loop to continuously improve the prover?


- Could none of the key intuitions from prior work be incorporated into a baseline?

**Clarity:**

As I mentioned before, the paper is not accessible for readers that are not aware of the theorem proving literature. The authors make an attempt to explain terminology such as "premise" and "tactic" but the example in Fig. 2 is not sufficient.

**Documentation:**

Yes

**Limitations:**

The paper does have an extensive limitation section in the Appendix.

**Opportunities For Improvement:**

- The paper is not accessible for readers that are not aware of the theorem proving literature. The authors make an attempt to explain terminology such as "premise" and "tactic" but the example in Fig. 2 is not sufficient.

- While I understand that the presented approach can't be directly compared to prior work, a baseline that incorporates at least one idea from prior work would be useful. Currently their learned system is compared to a non-ML system and a zero-shot model. Could none of the key intuitions from prior work be incorporated into a baseline?

- In similar vein, zero-shot seems like a weak baseline too. Could few-shot examples be provided to GPT3 or GPT4? Could the tactic generator fine-tuned model be replaced by few-shot GPT3 model (further reducing the training compute requirement albeit at the cost of inference compute)?

**Relation To Prior Work:**

As I mentioned before, the paper is not accessible for readers that are not aware of the theorem proving literature. The authors make an attempt to explain terminology such as "premise" and "tactic" but the example in Fig. 2 is not sufficient.

**Summary And Contributions:**

The paper presents a new dataset & benchmark: LeanDojo for theorem proving using Lean. From a dataset perspective, LeanDojo provides 97K theorems along with premise & tactic annotations that can be useful to train models. As a benchmark, it enables development and consistent evaluation of future methods by releasing the data and code publicly. They also present a retrieval-augment prover (Reprover) that retrieves relevant premises and then use these premises to generate the tactics at each step. Experimental results show that this approach can outperform heuristics and zero-shot GPT4 model, but the task of generalizing to novel premises still remains challenging.

---

> ### Author Response · Authors · 2023-08-20
> **Response to Reviewer crGr**
>
> Dear reviewer,
>
> Thank you for taking the time to review our paper! Your feedback is really valuable, and we address your questions/concerns below. Please feel free to follow up if you have further questions!
>
>
>
>
> ## Accessibility to Readers without Background in Theorem Proving
>
> We strive to make the paper accessible to general machine learning researchers, and we are sorry if the readability still needs improvements. We have updated the paper to make the example in Fig. 2 more salient and add a link to [LeanDojo’s extensive documentation](https://leandojo.readthedocs.io/).
>
> ## Additional Baselines Incorporating the Key Ideas in Prior Works
>
> Below we summarize the key ideas in prior works on learning to prove theorems in Lean:
>
> * Han et al., “Proof Artifact Co-training for Theorem Proving with Language Models”: Training the tactic generator jointly on tactic generation + auxiliary tasks is better than training on tactic generation alone.
>
> * Polu et al., “Formal Mathematics Statement Curriculum Learning”: The model can be improved further by using it to discover new proofs and adding the new proofs back to the training data. This procedure (called “expert iteration”) can be repeated multiple times.
>
> * Lample et al., "HyperTree Proof Search for Neural Theorem Proving": It may help to replace Best-First Search with HyperTree Proof Search (a search algorithm inspired by the Monte Carlo tree search in AlphaZero).
>
> These ideas are orthogonal to our work, as they can in principle be incorporated into either the baseline or our method. However, as explained at the end of Sec. 5, incorporating these ideas would make the method substantially more complex and computationally expensive.
>
> ## GPT-4 with Few-Shot Examples
>
> We conducted additional experiments to compare the zero-shot GPT-4 baseline with a few-shot GPT-4 baseline. Specifically, given a proof state, we use a finetuned model to retrieve k similar states and use them (together with their corresponding tactics) as few-shot examples for GPT-4 to generate tactics. We use k = 4, as increasing k did not help in small-scale preliminary experiments.
>
> We tested both models on 600 testing theorems from the “random” data split. They did not show a significant difference in performance (Pass@1 of 141/600 = 23.5% for zero-shot and 148/600 = 24.7% for few-shot). These experiments suggest GPT-4 might not be very good at generating tactics directly, even with few-shot examples in the prompt. A possible reason is that although there is Lean code in GPT-4’s training data, it does not contain latent information not visible in the code, e.g., intermediate proof states between tactics. We believe further research and more delicate methods are required to unleash GPT-4's potential for theorem proving.
>
>
> ## Are the Proofs Generated by the Model Guaranteed to Be Correct?
>
> Yes, they are guaranteed to be correct. Proof assistants such as Lean can check whether a proof correctly proves a given theorem statement. For Lean, the underlying principle is analogous to type checking. According to [Curry–Howard Isomorphism](https://en.wikipedia.org/wiki/Curry%E2%80%93Howard_correspondence), a theorem statement is a type, whereas a proof is a value. Therefore, proof checking corresponds to checking if the value has a given type. Other proof assistants may have different logical foundations, but the common takeaway is that proof checking is relatively easy in any proof assistant.
>
> ## Self-Training Loop to Continuously Improving the Prover
>
> Yes, this is feasible and has been adopted by prior works mentioned in a previous question (“expert iteration” in Polu et al. and “online training” in Lample et al.). We didn’t perform such online improvements, as it is orthogonal to our work and requires a substantially larger compute budget.

---

### Official Review · Reviewer_CZ3x · 2023-07-24
**Very good paper, but some issues need to be fixed/clarified**

**Rating:** 9
**Confidence:** 4

**Strengths:**

**Overall Strength:**
As mentioned in the *Summary and Contributions* section, this paper fills an essential gap in the literature by making research in the domain of formal theorem proving *accessible*. As the authors also state in their paper, there are several other papers that deal with automatic theorem proving but suffer from not open-sourcing the dataset/environment, model, or both. Unfortunately, it seems that for some papers, after they have been accepted, the code is not provided. This shows the importance of being up front with data, which the LeanDojo paper achieves.
(To illustrate the point above, consider the paper [1] (which the authors cite). Even though in that paper, the authors state in the Checklist (section 7) that "the code for the Equations environment will be open-sourced. We also plan to make our trained model publicly available to help people in the formal community", it seems this has not been followed through. I was not able to find the mentioned code/models, there seems to be no GitHub repository -at least not one that can be easily found- by any of the three main authors of that paper; and the code listed on the Openreview site of that paper [2] seems to point to existing Github repositories of Lean etc. rather then this code. I
[1] G. Lample et al., HyperTree Proof Search for Neural Theorem Proving, https://arxiv.org/pdf/2205.11491.pdf
[2] https://openreview.net/forum?id=J4pX8Q8cxHH )


**Other noteworthy points:**

- the paper consistently uses best practices (reproducibility - e.g., specifying exact commits on which the version of mathlib3 they base their paper on; training - hyperparameters, training setup, etc. are all carefully specified; comprehensive section on *Related Work*)

- the paper bases its environment on Lean 3, but also considers Lean4 (in the supplementary materials) and hints that Lean4Dojo will be made available for Lean4 as well. This is very important since progress on Lean4 and mathlib4 is developing rapidly, and it is expected that Lean4 will become the standard in the near future.

- issues in lean-gym are noticed, their root cause is identified (appendix A.2), and these are fixed by LeanDojo.

**Additional Feedback:**

I have considered the arXiv paper in addition to the pdf submitted here since I noticed that some errors in the version uploaded here are fixed in the version on arXiv: E.g., in Figure 1, as well as line 69, here is written incorrectly "mode_self," while the arXiv paper correctly refers to "mod_self".
I have not mentioned these issues in my review since I assume the authors will use (a perhaps updated version of) the current arXiv for the camera-ready version, which already includes the mentioned fixes.

I have given a score of **8 / clear accept**, *contingent* fixing the important issues I outlined in the *Opportunities For Improvement* section (in particular, 1) a detailed comparison between LeanStep and lean-gym needs to be made and 2) the git patch for Lean needs to be provided (or if it is provided already in the GitHub; though I haven't seen it, clearer instructions how I can find and apply the patch need to be given), so that the reader more clearly sees in what way LeanDojo is better than the competition and can easily reproduce the results.

Otherwise, I might have to **downgrade** after the reviewer-author discussion (but I hope that will not be the case).

But depending on how well and comprehensively these issues are addressed (there are some issues besides the ones I tagged in the *Opportunities For Improvement* section as important), I would also be happy to raise my score to a **9 / strong accept**, since this paper is already a lot of potential.

**Clarity:**

The paper is clearly written overall, but it's organization is complex since it consists of multiple parts (arXiv + GitHub + Zenodo + readthedocs). I have identified some specific issues below.

**Questions/Suggestions:**

- I am wondering why Clipala's work on Coq is cited in section 3; wouldn't citing the paper that introduces Lean would have been more relevant here? (Of course, Clipala's paper is important too; I'm simply confused why *only* Clipala was cited here.)

- I found myself frequently going back between Figures 1 and 2 while reading the paper. Perhaps these can be grouped together?

- lines 40-42 from the supplementary material: "Our modification is released as a Git patch that can be applied to any version of Lean 3 after March 24, 2022."
I wasn't able to find this patch under https://github.com/lean-dojo, nor any detailed instructions on how to apply it. Could you perhaps elaborate on this?
It would also be helpful to detail describe how comprehensive the changes that needed to be made to Lean3 were - and also whether you plan to carry out analogous changes for Lean4.

- In Appendix B, the LeanDojo Benchmark is described, which is an essential piece of the paper. It might be helpful to link to the dataset directly in the body of the text (e.g., in a footnote).
In a sequence of unlucky choices, I first was expecting to see a "LeanDojo Benchmark"  in the GitHub repository since I was initially looking in the GitHub repository for files you listed in the supplementary material, Appendix B.1 "LeanDojo Benchmark", as being part of the benchmark (`corpus.jsonl` etc.); not finding such a repository there, but discovering that there are Jupyter notebook files that generate the dataset I assumed that this dataset had to be generated. Only afterwards, I saw on the website that there are links to Zenodo mentioned in the supplementary material, but on Appendix B.3 and on the website as well.
While not every reader will proceed in such an unlucky way, also consider that a paper on arXiv will have a much longer half-life than a website that is dependent on a commercial provider (and continual author payment for the domain). It, therefore, might be helpful to not rely solely on the website to find quickly the source of the dataset and also link to Zenodo either it in the main body of the paper or add a reference in Appendix B.1, that in Appendix B.3 the dataset will be found.

- The above observation ties in with the fact that the organization of the LeanDojo project is rather complex since it consists of readthedocs + Github + Zenodo + arXiv. Since the main entry point is the website, as stated in supplementary section B3., I feel this is somewhat "fragile", since the existence of a website is not guaranteed to last; I would (also) make the main entry point the paper, in order to be future-proof for sure.
At the very least, I would add a small remark in the main body of the paper pointing to B.3, and clarifying that it is explained there how the project consists of different parts and how they are all tied together. Perhaps also include a quick summary in the paper of what further information can be found in readthedocs.This does not need to take anything away from the website, but it would be a layer of redundancy in case the website, as the current main entry point, goes down.

- I feel a lot of information is contained in the readthedocs.  It could be emphasized a bit more in the paper: Only in section B.3 and on the website it is mentioned that there are further documents. It would be a shame of all this information is somewhat hidden from the reader.

**Observations:**
The paper is well written, I didn't observe typos, and the explanations were mostly clear and detailed.

**Correctness:**

All verified claims in the paper seem correct. The dataset and benchmark are sounds.

**Documentation:**

The LeanDojo environment is well documented, and the specific commits of, e.g., the mathlib3 dataset, ProofNet dataset are mentioned from which the current dataset can be extracted. They also mention their modification of Lean 3, and Jupyter notebook files are provided to generate the dataset.

I haven't reproduced their dataset, but going over the code, which is well written and organized, I am confident I should be able to reproduce their dataset and benchmark with a reasonable amount of effort IF the patch they use to change Lean that resolves the premises' full names is provided / better documented.

As mentioned in the *Clarity* section, the organization of the various bits and pieces that make up LeanDojo is somewhat complex (readthedoc + Github + Zenodo + arxiv + website).

All other best practices surrounding dataset availability are present (license, hosting on Zenodo, etc.).

**Ethics:**

Not applicable.

**Limitations:**

**Questions/Suggestions:**
- The authors clearly indicated which mathlib3 commit / ProofNet commit uses to extract the data and that LeanDojo cannot extract those (somewhat rare) theorems that are defined using the keyword “def” (see line 169 from the appendix).
It would have been very helpful though, if the authors had given a precise number of how many such theorems actually are in that commit so that we don't have to rely on a somewhat vague adjective of "rare". Would it be possible to do that?

- LeanDojo doesn't support Lean proofs that are not based on tactics; although the authors state (line 254) that "tactic-style proofs [...] are sufﬁciently general since any proof can be converted to a tactic-style proof", which vindicates their approach.
Nonetheless, to ease the burden on the reader regarding the details of the Lean system, would it be possible to add a small paragraph or a reference that explains why (and how) a proof not based on tactics can be converted to one?

**Observations:**
- LeanDojo for Lean4 is only in alpha stage. But I am confident the authors will work towards the goal of advancing that codebase as well.
- data contamination when using GPT-4 for tactic generation (line 340) is acknowledged
- appendix C.3 contains a detailed and convincing account of why a comparison with LLMs is not meaningful
- For Lean 4, LeanDojo Benchmark 4 is generated, but ReProver is not trained because a dataset split cannot be performed since premise extraction is not yet implemented.

(Lastly, I have some doubts about whether a premise selection approach will ever be able to be scaled to the point where mathematical statements --formalized in Lean-- that are currently unproven statements on the frontier of mathematical research will generate a proof; but this is not a criticism of this paper, but rather of the general research direction.)

**Opportunities For Improvement:**

**Important issues**

1) A more comprehensive comparison of LeanDojo to LeanStep and lean-gym would have been very helpful. The authors obviously spend quite some time familiarizing themselves with these systems (e.g., they found errors in how lean-gym works). Since these build on Lean, which is already complex by itself, it would be a great service to the readers to include a description comparing these systems, that goes in detail. In is clearly explained how LeanDojo collects the prooftree - but how does lean-gym do it? For example, LeanDojo works on mathlib3, while lean-gym works on Lean directly (the paper on lean-gym [lean-gym] says: "To solve these issues we implemented lean-gym1 – a simple REPL interface over the standard input/output implemented in Lean directly").
Such explanation regarding similarities and differences in implementation, inner workings etc. (ideally summarized in a table) are important if one wants to build on these environments since it illustrates how easy/difficult it is to get them to work, how strongly they are coupled to a specific version of Lean etc.

[lean-gym] S. Polu et al., Formal Mathematics Statement Curriculum Learning, https://arxiv.org/pdf/2202.01344.pdf

2) Clearly indicate the patch which was used to resolve the full names of the premises from Lean3 (and provide installation instructions).

**Less important issues**

1) A slightly computer science-y view of mathematics:
 lines 35-38: "Formal proofs are essentially computer programs. But unlike conventional programs in C++ or Python, the correctness of proofs can be verified easily using proof assistants. Therefore, theorem proving is a special form
of code generation, with rigorous evaluation and no room for the model to hallucinate."
When *verifying* proofs one may think of theorem proving as a special form of code generation. But when it comes to *discovering*, no professional mathematician would say his work is "code generation", since proving a theorem involves intuition, visualization etc. I would perhaps slightly rewrite it to clarify this.
Generally, it seems that a mathematical perspective could improve this well-written paper even more.

2) I was missing an analysis of which mathematical areas the theorem from the LeanDojo Benchmark belong to (on line 220 is just mentioned "analysis, algebra, and geometry" which is a somewhat vague description; at least a somewhat more finegrained-description would have been very helpful here; or perhaps some illuminating examples of theorems, so that the reader can assess their difficulty, mathematical domain etc.). So far the paper is interesting more for computer scientists, than for mathematicians.

3) Issues mentioned in other sections:
 see my questions from the "Limitations" section, as well as the "Clarity" section.

**Relation To Prior Work:**

The paper comprehensively discusses previous work and how it relates to previous Lean environments. One thing that I would have found helpful would have been a more detailed comparison of LeanDojo to LeanStep and lean-gym; I have mentioned this in the *Opportunities For Improvement* section.

**Summary And Contributions:**

A new, open-source environment for extracting proof trees out of Lean3 / mathlib3 is presented. This environment is called LeanDojo, and it is tailored to facilitate training data for a machine-learning approach. This is performed in the following way:

- it extracts the entire proof tree (root is the theorem to be proved and the leaf the state after all proof obligations have been discharged)
- it locates the premises that were used in the proof

The LeanDojo Benchmark is constructed in this sense, consisting of a diverse area of mathematics and a Retrieval-Augmented Theorem Prover, called ReProver, is trained. Their theorem prover performs well on the premise selection task, as well as on the theorem proving task. In particular, it outperforms GPT-4 in terms of tactic generation.

While the theorem prover is a nice addition, the main contribution is the LeanDojo environment and the improvements it brings over existing environment. This is essential progress towards opening up automatic theorem proving in formal mathematics, as issues with the previous two environments, LeanStep, and lean-gym (both of which are discussed in the paper), are fixed. I expect that important work will be built upon LeanDojo. The thoughtful execution, documentation, and user-friendly presentation of LeanDojo will (hopefully) serve as a standard for environments related to the other well-known theorem provers (Mizar, Coq etc.).

---

> ### Author Response · Authors · 2023-08-20
> **Response to Reviewer CZ3x (1/2)**
>
> Dear reviewer,
>
> Thank you for the detailed review! We truly appreciate your in-depth comments and constructive feedback. Below we address your questions and concerns. Please feel free to follow up if you have further questions!
>
>
> ## Detailed Comparison between LeanDojo and LeanStep/lean-gym
>
> The updated appendix contains a new subsection (B.3) and a new table (A) dedicated to this comparison. We also outline it here:
>
> ### Functionality
>
> LeanDojo supports both data extraction and interacting with Lean programmatically. In contrast, LeanStep is only for data extraction, and lean-gym is only for interacting with Lean. They are not actively maintained, so they do not support recent versions of mathlib (tested on August 11, 2023, using mathlib commit [32a7e535287f9c73f2e4d2aef306a39190f0b504](https://github.com/leanprover-community/mathlib/tree/32a7e535287f9c73f2e4d2aef306a39190f0b504)). Also, neither of them supports Lean 4. Our LeanDojo fully supports recent mathlib and Lean 4. Furthermore, LeanStep cannot extract premise information and is not applicable to repos other than mathlib. Last, LeanDojo comes with comprehensive documentation and unit tests, whereas other tools barely have any.
>
>
> ### Implementation details
>
> LeanStep and LeanDojo use different mechanisms to extract ASTs and proof trees. LeanStep implements an [ad-hoc parser](https://github.com/jasonrute/lean_proof_recording/blob/master/lean_proof_recording/parser.py) in Python for parsing Lean code into ASTs. It also [intercepts Lean's tactic system to insert logging code](https://github.com/jasonrute/lean_proof_recording/blob/master/lean_modifications/tactic_modifications.lean). Then the logs are used to reconstruct proof trees. This implementation is brittle and does not work for the current versions of Lean/mathlib. In contrast, LeanDojo relies on Lean’s built-in mechanisms for exporting ASTs and proof states (`lean --ast --tsast --tspp`), which works robustly for recent Lean/mathlib. This mechanism was developed after LeanStep.
>
> Regarding interaction with Lean, both lean-gym and LeanDojo rely on Lean's metaprogramming APIs, and LeanDojo partially builds upon lean-gym's code. However, lean-gym has a critical issue that it misjudges many correct proofs as incorrect (Appendix B.2). The main reason is that lean-gym fails to distinguish two subtly different cases when constructing the proof environment: (1) opening a namespace; (2) being inside a namespace. LeanDojo does not suffer from this issue. Instead of operating as a standalone program in the IO monad, it wraps the interaction code into [a special tactic](https://github.com/lean-dojo/LeanDojo/blob/32c14d5839e26be80ebff34190dcecd812eeb30f/src/lean_dojo/interaction/lean3_repl.lean#L564), which is inserted into the correct location in the proof. Therefore, the interaction code is guaranteed to run in the same environment as the original human-written proof.
>
>
> ## The Patch for Extracting Premises in Lean 3
>
> The patch is publicly available [in the LeanDojo repo](https://github.com/lean-dojo/LeanDojo/blob/main/src/lean_dojo/data_extraction/0001-Modify-Lean-for-proof-recording.patch). However, it is an implementation detail hidden from users and does not need to be installed. As in [Getting Started](https://leandojo.readthedocs.io/en/latest/getting-started.html), the user simply specifies a Lean repo (e.g., https://github.com/zhangir-azerbayev/ProofNet, `876bf5f9a424e92fc74d7e72c0bee0eb77bdc0b1`). Under the hood, LeanDojo automatically identifies the required Lean version and applies the patch. This works out of the box without the user being aware of the patch. We have updated the paper to clarify. Sorry for any confusion.
>
> In addition, the patch is only required for Lean 3. For Lean 4, the elaborator can be accessed through metaprogramming in Lean itself, which does not require modifying Lean’s C++ code. The core of our Lean 4 implementation is [here](https://github.com/lean-dojo/LeanDojo/blob/main/src/lean_dojo/data_extraction/ExtractData.lean).
>
>
> ## Lean 4 Support
>
> We have upgraded LeanDojo to fully support Lean 4 (no longer in the alpha stage), including extracting premise information. Please see our common response at the top of this page.

---

> > ### Author Response · Authors · 2023-08-20
> > **Response to Reviewer CZ3x (2/2)**
> >
> > ## Theorem Proving vs. Code Generation
> >
> > We agree that the original statement (`Formal proofs are essentially computer programs. ... theorem proving is a special form of code generation, ...`) is simplistic and can be misleading. In the updated version, we have rephrased it as: `From a computer science perspective, formal proofs can be treated as programs~\cite{howard1980formulae}. … theorem proving may be considered a special form of code generation, …`, where the citation refers to the Curry-Howard isomorphism (the relationship between programs and proofs that sits behind the logical foundation of Coq and Lean).
> >
> > ```bibtex
> > @article{howard1980formulae,
> >   title={The formulae-as-types notion of construction},
> >   author={Howard, William A},
> >   journal={To HB Curry: Essays on Combinatory Logic, Lambda Calculus and Formalism},
> >   year={1980}
> > }
> > ```
> >
> > ## Details of Mathematical Areas Covered by LeanDojo Benchmark
> >
> > We updated the paper to link to pages ([1](https://leanprover-community.github.io/mathlib_stats.html), [2](https://eric-wieser.github.io/mathlib-import-graph/)) containing statistics, visualizations, and examples of mathematics covered by mathlib3. They will remain relatively stable, as the development of mathlib3 has been [frozen since June 22, 2023](https://leanprover.zulipchat.com/#narrow/stream/113486-announce/topic/Freeze.20of.20mathlib.203), and we have updated the paper to use a more recent version ([32a7e535287f9c73f2e4d2aef306a39190f0b504](https://github.com/leanprover-community/mathlib/tree/32a7e535287f9c73f2e4d2aef306a39190f0b504) released on August 5, 2023)
> >
> >
> >
> >
> > ## How Many Theorems Are Defined Using the `def` Keyword?
> >
> > It is **zero** in mathlib (for both Lean 3 and Lean 4). Mathlib has a linter for detecting such unidiomatic uses (see [1](https://github.com/leanprover-community/mathlib/blob/32a7e535287f9c73f2e4d2aef306a39190f0b504/src/tactic/lint/misc.lean#L246) and [2](https://github.com/leanprover/std4/blob/dbffa8cb31b0c51b151453c4ff8f00ede2a84ed8/Std/Tactic/Lint/Misc.lean#L107)). The linter is a part of mathlib’s CI pipeline that runs automatically for each commit. Therefore, in any commit passing all CI checks, there shouldn’t be any theorem defined using the `def` keyword.
> >
> >
> >
> >
> > ## How to Convert Proofs into Tactic-Style Proofs?
> >
> > We have updated the paper with a new footnote: “Another common type of proofs is ‘term-style proofs’. Any term-style proof `X` can always be converted into an equivalent tactic-style proof `exact X`, though such conversion may lead to unidiomatic proofs.”
> >
> >
> > ## Why is Clipala's Book on Coq Cited in Sec. 3?
> >
> > We intended to cite a book on functional programming with dependent types. We agree that other options might be more relevant to Lean, and we have updated the paper with David Thrane Christiansen’s book [Functional Programming in Lean](https://leanprover.github.io/functional_programming_in_lean/).
> >
> >
> > ## Link Everything in the Paper Instead of Relying on the Website
> >
> > Thank you for the great suggestion! We have updated the appendix with a new section (A) containing links to assets introduced by our work. It includes:
> >
> > * LeanDojo's codebase for data extraction and interaction with Lean: https://github.com/lean-dojo/LeanDojo
> > * LeanDojo's documentation: https://leandojo.readthedocs.io
> > * Datasets: (1) [LeanDojo Benchmark](https://zenodo.org/record/8242196) with DOI `10.5281/zenodo.8242196`. (2) [LeanDojo Benchmark 4](https://zenodo.org/record/8242200) with DOI `10.5281/zenodo.8242200`.
> > * ReProver's code and models: https://github.com/lean-dojo/ReProver
> > * ChatGPT plugin: https://github.com/lean-dojo/LeanDojoChatGPT
> > * LeanDojo Website: https://leandojo.org/

---

> > > ### Comment · Reviewer_CZ3x · 2023-08-21
> > >
> > > Thank you for your detailed response, for clarifying a large number of issues, and for updating LeanDojo to version 4; great work! I will raise my score, but I will nonetheless await your further response before I do so.
> > >
> > > - Regarding lean-gym and LeanStep: As you mentioned, LeanDojo surpasses both of these. Is there something (in terms of data extraction or interaction) that these can do that LeanDojo cannot do? (In other words, is LeanDojo a strict improvement over them in all aspects?)
> > >
> > > - you mentioned you support https://github.com/zhangir-azerbayev/ProofNet in order to automatically identify Lean's version. Have you hardcoded this repo's structure to allow it to identify Lean's version? I am not sure how to make sense of "_Under the hood, LeanDojo automatically identifies the required Lean version_". In the *Getting Started* section, the "hello word" repo link you give, https://github.com/yangky11/lean-example, uses a different format.
> > > (If yes, which repos do you support? Or what layout do the repos have to adhere to? Do you support miniF2F as well?)
> > >
> > > - You mentioned that: "_Another common type of proofs is ‘term-style proofs’. Any term-style proof X can always be converted into an equivalent tactic-style proof exact X, though such conversion may lead to unidiomatic proofs._"
> > > I believe term-style proofs and tactic-style proofs are the only major types of proofs possible in Lean, but I believe there are also proofs like calculation proofs, starting with `calc` (though these form a minority). LeanDojo does not cover calculation proofs?
> > > If that is the case, please state clearly what types of proofs LeanDojo covers, and which it doesn't (the paper is still great, even if not everything is covered).

---

> > > > ### Author Response · Authors · 2023-08-21
> > > > **Response to Reviewer CZ3x**
> > > >
> > > > We are happy to address your questions and really appreciate your willingness to raise the score!
> > > >
> > > >
> > > > ## Is There Something LeanStep/lean-gym Can Do That LeanDojo Cannot?
> > > >
> > > > lean-gym has a [shrink_proof](https://github.com/openai/lean-gym/blob/1585ac4d2e56a1ceb72243ce859645b9d0069d34/src/repl.lean#L327) function undocumented in their paper (Polu et al., ICLR 2023). From [comments in their code](https://github.com/openai/lean-gym/blob/1585ac4d2e56a1ceb72243ce859645b9d0069d34/src/tools/shrink_proof.lean#L12%E2%80%93L26), shrink_proof removes redundant tactics in a specific form of proofs. We don’t have sufficient information on the context in which shrink_proof is useful. Other than that, LeanDojo supports everything lean-gym can do.
> > > >
> > > > To our knowledge, LeanDojo’s extracted data contains all information available in LeanStep.
> > > >
> > > >
> > > >
> > > >
> > > > ## How to Get the Required Lean Version of a Repo in Lean 3?
> > > >
> > > >
> > > > All repos in Lean 3 have a configuration file named leanpkg.toml containing the required Lean version. For example, here are the leanpkg.toml for [mathlib](https://github.com/leanprover-community/mathlib/blob/32a7e535287f9c73f2e4d2aef306a39190f0b504/leanpkg.toml#L4), [ProofNet](https://github.com/zhangir-azerbayev/ProofNet/blob/96c8978850fba27a748c941c276d9e178d0efbd4/leanpkg.toml#L4), [miniF2F](https://github.com/facebookresearch/miniF2F/blob/5271ddec788677c815cf818a06f368ef6498a106/leanpkg.toml#L4), and [lean-example](https://github.com/yangky11/lean-example/blob/5a0360e49946815cb53132638ccdd46fb1859e2a/leanpkg.toml#L4).
> > > >
> > > >
> > > > ## Clarification on Proof Styles
> > > >
> > > > Most proofs in mathlib are term-style, tactic-style, or a mix of them (example [here](https://leanprover.github.io/theorem_proving_in_lean4/tactics.html#structuring-tactic-proofs)). Other styles include [calculational proofs](https://leanprover.github.io/theorem_proving_in_lean4/quantifiers_and_equality.html#calculational-proofs) and [the conversion tactic mode](https://leanprover.github.io/theorem_proving_in_lean4/conv.html), but they are relatively rare. Regardless of the proof style, ultimately the proof is converted into a term and then type-checked by Lean’s kernel. In this sense, term style is the most general proof style. In principle, you can write down the proof term directly. As we argued in the paper, tactic style is as general as term style, since any term-style proof can be converted into an equivalent tactic-style proof. In contrast, calculational proofs and conversion tactic mode are only applicable to certain types of theorems.
> > > >
> > > > LeanDojo only supports tactic-style proofs. When using LeanDojo to prove a theorem, the model must generate a tactic-style proof. Nevertheless, that theorem may have a human-written proof in any style.

---

### Author Response · Authors · 2023-08-20
**Thank you to all reviewers and meta-reviewers!**

Dear reviewers and meta-reviewers,

We appreciate your time and effort in engaging with our work and providing constructive feedback! We have received five very positive and thoughtful reviews. We are particularly excited that reviewers have found LeanDojo an essential progress towards automated theorem proving and potentially a standard environment for many important future works to build upon.

We respond to each reviewer to address their questions and comments. The paper and supplementary PDFs have been updated with suggested revisions. We welcome any follow-up discussions!


## Update on Lean 4 Support



At the time of submission, Lean 3 was the stable version, and Lean 4 was experimental. Accordingly, LeanDojo fully supported Lean 3 but was experimental for Lean 4, without the capability of extracting premise information for Lean 4. However, after the submission, the Lean/mathlib community has made substantial progress in migrating mathlib from Lean 3 to Lean 4. Now, Lean 4 has become the stable version. The mathlib for Lean 3 has been frozen since June 22, and all future developments will happen in Lean 4.


During this time, we also finished upgrading LeanDojo to fully support Lean 4, including extracting premise information. Therefore, we have re-generated the two datasets in this paper:

-   LeanDojo Benchmark: 98,641 theorems/proofs, 217,639 tactics, and 130,151 premises extracted from mathlib (commit [32a7e535287f9c73f2e4d2aef306a39190f0b504](https://github.com/leanprover-community/mathlib/tree/32a7e535287f9c73f2e4d2aef306a39190f0b504) released on Aug. 5)
-   LeanDojo Benchmark 4: 100,780 theorems/proofs, 209,133 tactics, and 101,500 premises extracted from mathlib4 (commit [355541ae7a2455222f179dcf7f074aa2c45eb8aa](https://github.com/leanprover-community/mathlib4/tree/355541ae7a2455222f179dcf7f074aa2c45eb8aa) released on Aug. 10).

We have re-run experiments on the new datasets and updated Table 1, 2, B, and C. All conclusions drawn from these tables remain the same.

---

### Decision · Program_Chairs · 2023-09-22

**Decision:**

Accept (Oral)

**Comment:**

The paper introduces an open-source environment (LeanDojo) to extract proof trees out of Lean3, together with the premises of those proofs. These exacted proof trees are then used to build a new dataset for building stronger theorem provers.
Finally, this dataset is used to train a sota retrieval augmented automated theorem proving model (ReProver) which performs premise retrieval. This model outperforms GPT-4 on tactic generation, and also finds proofs to theorems in the library that are missing proofs. This model also uses significantly less compute that previous models i.e. one GPU for 5 days.

Strengths:
1. This paper has overwhelming support from all reviewers.
2. The paper fixes issues with previous environments LeanStep and Lean-gym and advances the sota quite considerably in this area.
3. The paper execution is thoughtful with excellent and user-friendly documentation, and uses the very best practices.
4. Unlike previous works with empty claims to be open source, this paper is actually already open-source, and will help the community make large strides of progress.

Weaknesses:
1. The only main weakness is that the paper can be hard to understand for folks outside the area of theorem proving - but the authors have taken efforts to address this in the rebuttal.